# Light-induced giant enhancement of nonreciprocal transport at KTaO$_3$-based interfaces

Xu Zhang[1,8], Tongshuai Zhu[2,3,8], Shuai Zhang [ORCID][2,8], Zhongqiang Chen[1], Anke Song[1], Chong Zhang[1], Rongzheng Gao[1], Wei Niu [ORCID][1], Yequan Chen[1], Fucong Fei [ORCID][2], Yilin Tai[4], Guoan Li[5], Binghui Ge [ORCID][4], Wenkai Lou [ORCID][6], Jie Shen [ORCID][5], Haijun Zhang [ORCID][2], Kai Chang [ORCID][6], Fengqi Song [ORCID][2] ✉, Rong Zhang [ORCID][1,7] ✉ & Xuefeng Wang [ORCID][1] ✉

Nonlinear transport is a unique functionality of noncentrosymmetric systems, which reflects profound physics, such as spin-orbit interaction, superconductivity and band geometry. However, it remains highly challenging to enhance the nonreciprocal transport for promising rectification devices. Here, we observe a light-induced giant enhancement of nonreciprocal transport at the superconducting and epitaxial CaZrO$_3$/KTaO$_3$ (111) interfaces. The nonreciprocal transport coefficient undergoes a giant increase with three orders of magnitude up to $10^5 \, \text{A}^{-1} \, \text{T}^{-1}$. Furthermore, a strong Rashba spin-orbit coupling effective field of 14.7 T is achieved with abundant high-mobility photocarriers under ultraviolet illumination, which accounts for the giant enhancement of nonreciprocal transport coefficient. Our first-principles calculations further disclose the stronger Rashba spin-orbit coupling strength and the longer relaxation time in the photocarrier excitation process, bridging the light-property quantitative relationship. Our work provides an alternative pathway to boost nonreciprocal transport in noncentrosymmetric systems and facilitates the promising applications in opto-rectification devices and spin-orbitronic devices.

Recently, nonlinear transport has been regarded as a sophisticated probe of the symmetry breaking in noncentrosymmetric materials/heterostructures[1]. Inversion symmetry breaking changes the original electronic band structure, thus generating a myriad of intriguing physical properties, such as nonlinear Hall effect[2], quantum-metric-induced nonlinear transport[3,4], bulk photovoltaic effect[5], circular photogalvanic effect[6], and helicity dependent terahertz emission[7]. Another striking example is the nonreciprocal charge transport with both spatial inversion symmetry breaking and time reversal symmetry breaking[8]. In this case, the longitudinal resistance exhibits a

[1]Jiangsu Provincial Key Laboratory of Advanced Photonic and Electronic Materials, State Key Laboratory of Spintronics Devices and Technologies, School of Electronic Science and Engineering, Collaborative Innovation Center of Advanced Microstructures, Nanjing University, Nanjing 210093, China. [2]National Laboratory of Solid State Microstructures, School of Physics, Nanjing University, Nanjing 210093, China. [3]College of Science, China University of Petroleum (East China), Qingdao 266580, China. [4]Information Materials and Intelligent Sensing Laboratory of Anhui Province, Institutes of Physical Science and Information Technology, Anhui University, Hefei 230601, China. [5]Beijing National Laboratory for Condensed Matter Physics and Institute of Physics, Chinese Academy of Sciences, Beijing 100190, China. [6]State Key Laboratory for Superlattices and Microstructures, Institute of Semiconductors, Chinese Academy of Sciences, Beijing 100083, China. [7]Department of Physics, Xiamen University, Xiamen 361005, China. [8]These authors contributed equally: Xu Zhang, Tongshuai Zhu, Shuai Zhang. ✉e-mail: songfengqi@nju.edu.cn; rzhang@nju.edu.cn; xfwang@nju.edu.cn

unique directional response depending on the current and magnetic field directions. A phenomenological expression for this directional transport is expressed by generalizing Onsager's theorem as follows[9,10]

$$R(B, I) = R_0 \left(1 + \beta B^2 + \gamma BI\right) \quad (1)$$

where $R_0$, $\beta$, $\gamma$, $I$, and $B$ represent the resistance at zero magnetic field, the coefficient of the normal magnetoresistance (MR), the nonreciprocal transport coefficient, electric current, and the external in-plane magnetic field perpendicular to $I$, respectively. Experimentally, non-reciprocal charge transport has been demonstrated in a variety of low-symmetry systems, including topological insulators/semimetals[11–14], polar semiconductors[15,16], semiconductor heterostructures[17], superconducting systems[18–21] and correlated oxide heterostructures[22–24]. Achieving a high magnitude of this directional charge transport is of vital importance towards the next-generation two-terminal rectification and spin-orbitronic devices[20,25]. To date, considerable efforts have been devoted to achieving the larger $\gamma$ value via gate bias[17,22] and ionic-liquid gating[26]. However, these electric-field-effect gating means remain highly limited to manipulate the Rashba spin-orbit coupling (SOC) strength and the Fermi level due to the relatively weak carrier tunability[22], thus seriously restricting the $\gamma$ enhancement.

Optical control is a paramount tool to explore emergent quantum phenomena in quantum materials[27]. In addition to ultrafast laser pulses, conventional light source has also been proved to be a powerful tool to manipulate quantum phases, such as light-induced superconductivity in an organic Mott insulator[28], light-induced insulator-metal transition in an ultrathin $SrRuO_{3-\delta}$[29], and optical modulation of Rashba SOC in two-dimensional electron gases (2DEGs) at $SrTiO_3$-based interfaces[30]. It is highly desirable to unveil the sophisticated interplay between light and nonreciprocal transport. In this regard, however, light-induced nonreciprocal transport in quantum materials has remained largely unexplored yet.

As a $5d$ transition-metal oxide, $KTaO_3$ (KTO)-based interfaces hosting 2DEGs have aroused considerable interest owing to fascinating physical properties, such as 2D anisotropic superconductivity[31–34], tunable strong Rashba SOC[35–39], and robust spin polarization[40]. Especially, the superconductivity at KTO-based interfaces is strongly dependent on the crystalline orientation of KTO with the highest superconducting transition temperature ($T_C$) of 2.2 K at (111)-oriented surface[31], very different from $SrTiO_3$-based counterparts, indicating the underlying rich physics in KTO. Furthermore, the interfacial superconductivity can be further manipulated by the electric-field with/without ionic-liquid gating, unveiling the electron-doped surface[41]. Specifically, KTO-based 2DEGs are also very sensitive to the conventional light source[35,42,43], providing an ideal platform to explore the complicated interplay between photoelectrons and heavy $5d$ electrons, thus readily tuning the Rashba SOC strength in a broader range under light illumination. However, the nonreciprocal charge transport via optical modulation in the KTO-based 2DEGs still remains elusive.

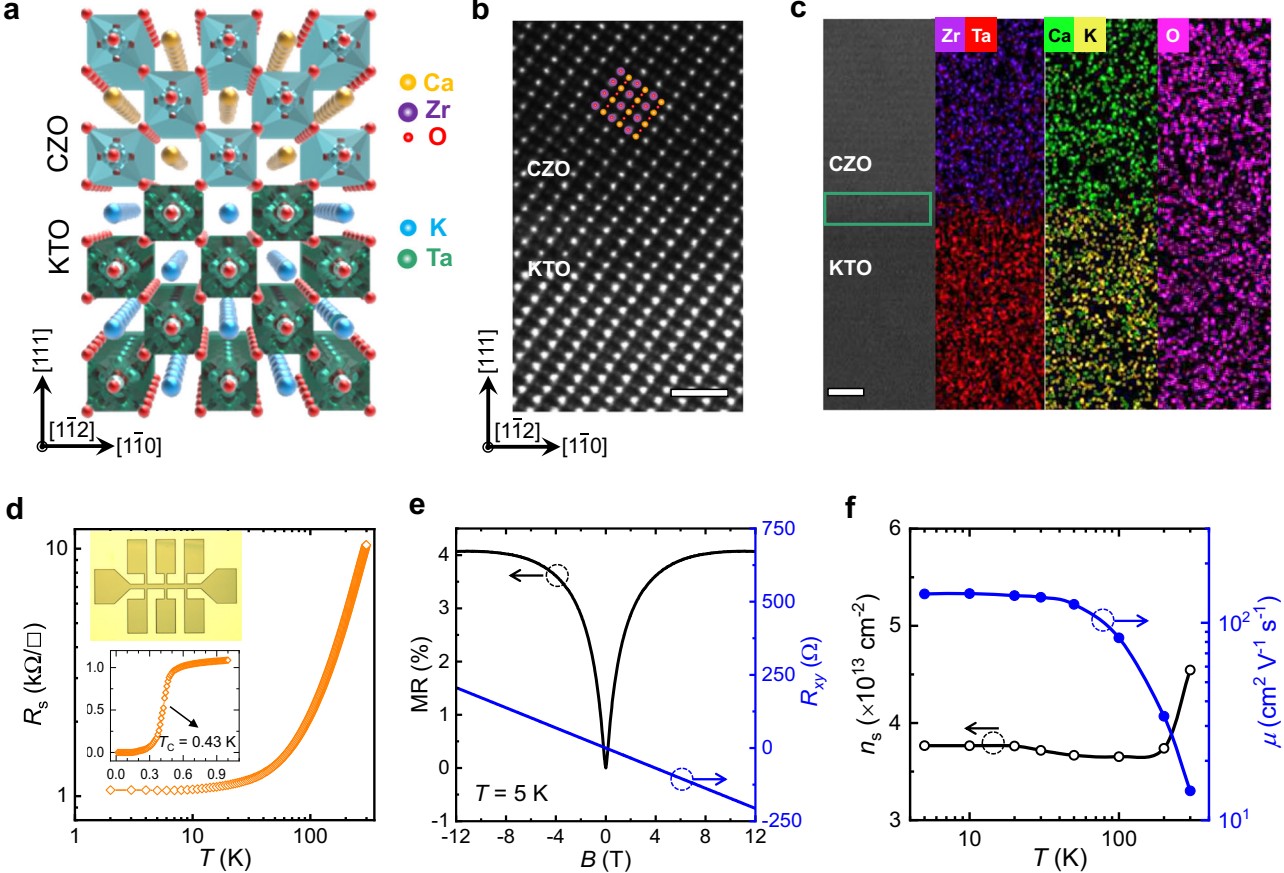

**Fig. 1 | Basic structural and transport properties. a** Schematic illustration of the crystal structure for the perovskite heterostructures of CZO/KTO. **b** STEM-HAADF image of CZO/KTO, overlapped with the atomic configuration. The scale bar is 1 nm. **c** STEM image and the corresponding EDX elemental mapping at the interface. The green box indicates the area where 2DEGs are located at the interface. The scale bar is 5 nm. **d** Temperature-dependent sheet resistance ($R_s$) of the CZO/KTO heterostructure. The upper inset shows the optical image of the Hall-bar device. The lower inset indicates the superconductivity observed at cryogenic temperature. **e** MR and Hall resistance ($R_{xy}$) with respect to the applied out-of-plane magnetic field at 5 K. **f** Temperature-dependent carrier density ($n_s$) and mobility ($\mu$) of the device without illumination.

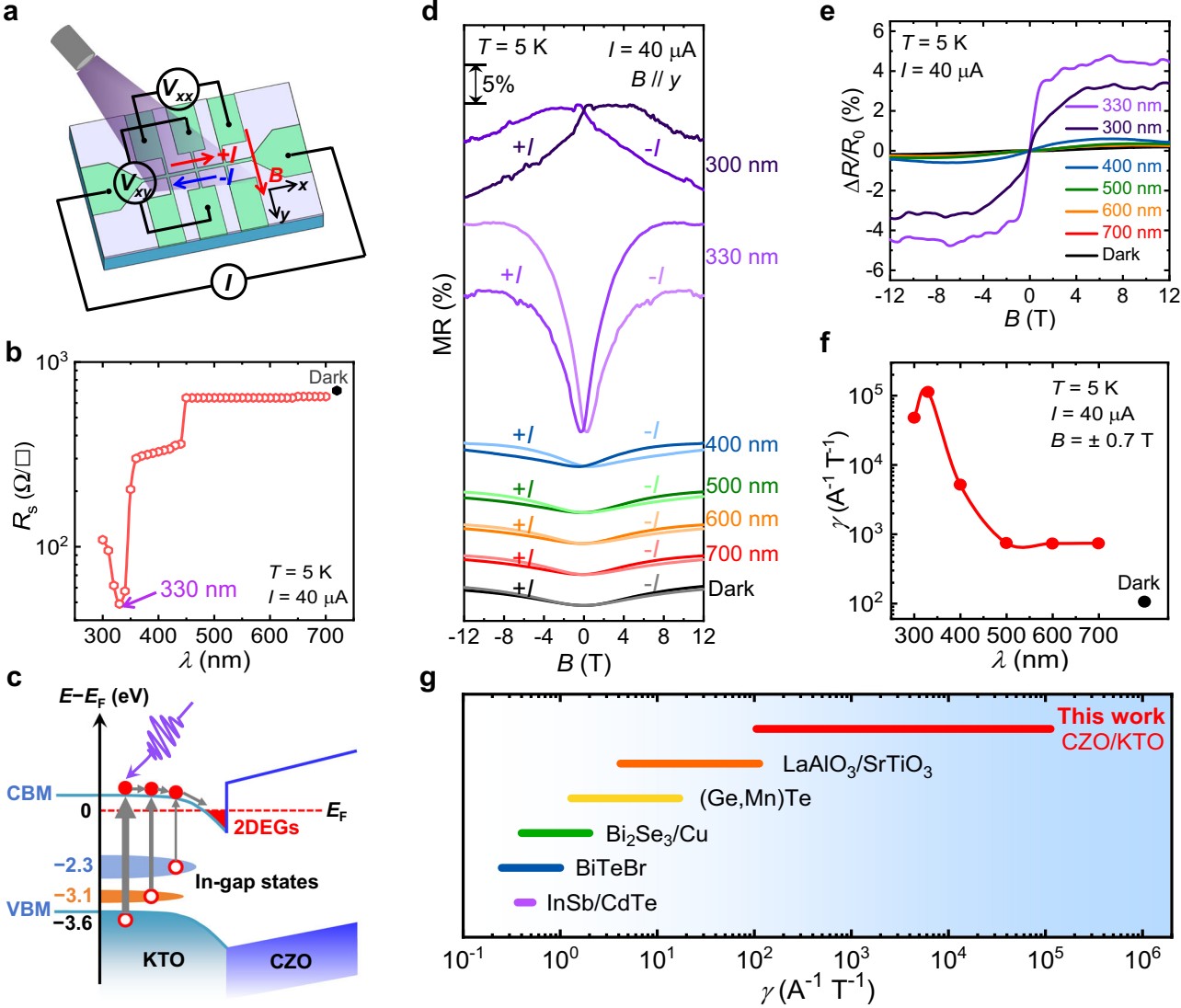

**Fig. 2 | Light-induced giant enhancement of nonreciprocal transport.**
**a** Schematic measurement configuration diagram of the optical modulation for the Hall-bar device of 2DEGs at the CZO/KTO heterointerface. Note that the in-plane magnetic field is perpendicular to the current direction. **b** Photoinduced change of the resistance of 2DEGs as a function of the incident light wavelength at 5 K, showing that the irradiation at 330 nm with the energy (-3.76 eV) slightly exceeding the optical bandgap of KTO leads to the largest photoconductance. **c** Schematic band structure diagram of the light-gating mechanism at the CZO/KTO interface. The thicker grey arrows denote the more photoexcited electrons. **d** In-plane MR curves at various wavelengths. The measurements are performed for both $I = \pm40\,\mu A$ at 5 K. The MR curves are shifted vertically for clarity. **e** The ratio of the resistance change ($\Delta R/R_0$ extracted from **d**) as a function of magnetic field at various wavelengths. **f** Wavelength-dependent nonreciprocal transport coefficient extracted from (**e**). The fitting range of the magnetic field is set to be $\pm0.7\,T$. The corresponding curves and parameters of the dark condition are also included in (**b**,**d**–**f**) for comparison. **g** The summarized graph for $\gamma$ of various material systems, such as InSb/CdTe[17], BiTeBr[26], Bi$_2$Se$_3$/Cu[54], (Ge,Mn)Te[16] and LaAlO$_3$/SrTiO$_3$[22], that utilize various methods for the nonreciprocal transport modulation.

In this article, we demonstrate a light-induced giant enhancement of nonreciprocal transport at superconducting and epitaxial CaZrO$_3$/KTO (111) (CZO/KTO) heterostructures grown by pulsed laser deposition (PLD). The magnitude of the nonreciprocal transport coefficient ($\gamma$) undergoes a huge enhancement of 1000 times to $10^5\,A^{-1}\,T^{-1}$ when illuminated at 330 nm. The Rashba SOC strength also experiences a remarkable increase, which is accompanied by the prominent photocarriers excited from the valence band of KTO. Combining first-principles calculations and data fitting, the giant $\gamma$ enhancement is attributed to the larger Rashba coefficient and the longer carrier relaxation time under ultraviolet light illumination. Our work not only demonstrates the interplay between light and nonreciprocal transport but also provides great potential for the realization of emergent two-terminal rectification devices and spin-orbitronic devices.

## Results

### Formation of high-quality 2DEGs at the CZO/KTO heterointerfaces

The CZO/KTO heterostructured films are prepared by depositing CZO films on (111)-oriented KTO single-crystal substrates by the PLD (see "Methods" section for details). Figure 1a schematically shows the typical perovskite crystal structure for CZO/KTO heterostructures. The lattice parameter of CZO is very close to that of KTO with a lattice mismatch of only -0.6%. This is beneficial to a layer-by-layer epitaxial growth of the CZO/KTO perovskite heterostructures. Such a perfect epitaxy is indicated from the coincident diffraction peak of CZO from x-ray diffraction (XRD) (Supplementary Fig. 1a). The microstructure of the heterointerface is further examined by high-resolution scanning transmission electron microscopy (STEM) in a high angle annular dark field (HAADF) mode. From the STEM-HAADF image (Fig. 1b), it can be

seen that the homogeneous and crystalline phase of CZO thin film is perfectly grown on the KTO (111) substrate. The enlarged atomic configuration overlapping on the STEM image is shown in Supplementary Fig. 1b. The corresponding energy dispersive x-ray spectroscopy (EDX) elemental mapping (Fig. 1c) indicates the clear and smooth interface between the KTO substrate and the CZO film without elemental interdiffusion. Note that the noticeable Ca element in the KTO substrate and K element in the CZO film (Fig. 1c) are actually artifacts due to the very close energy edges of Ca and K from EDX (Supplementary Fig. 2).

The transport properties are measured with a Hall-bar configuration (upper inset of Fig. 1d). The detailed fabrication process of the Hall-bar device is presented in Supplementary Fig. 3. We observe a typically metallic behavior in the whole temperature range (Fig. 1d), indicating the formation of 2DEGs at their heterointerface. This is ascribed to the formation of oxygen vacancies at the surface of KTO[44]. The homogeneous conducting channel is evidenced by the arbitrarily selected multi-electrode measurements (Supplementary Fig. 4a). The anisotropic MR further reveals its 2D conducting characteristic with the thickness of around 4.35 nm (Supplementary Fig. 4b). Remarkably, the sheet resistance ($R_s$) undergoes a rapid transition to a zero-resistance state when the temperature further decreases (see the lower inset of Fig. 1d), indicating the occurrence of superconductivity at the interface. The $T_C$ is determined to be 0.43 K, which is defined by the temperature corresponding to 50% of the resistance according to the previous work[31,32,44]. Its $T_C$ is much lower than those of non-epitaxial KTO-based heterointerfaces[31,32], but comparable to that of epitaxial LaVO$_3$/KTO (111) system (~0.5 K)[45]. Figure 1e depicts the field-dependent MR (MR = $\frac{R(B)-R_0}{R_0} \times 100\%$) and Hall resistance ($R_{xy}$) at 5 K. The MR rapidly decreases to a minimum, forming a cusp shape at the low field and manifesting a notable characteristic of weak antilocalization (WAL). The emergence of the WAL effect implies the non-negligible contribution of the Rashba SOC[37,46]. Besides, the linear behavior of $R_{xy}$ with a negative slope indicates that the charge carriers are electrons and only one type of carriers dominates the transport. The temperature-dependent carrier density ($n_s$) and the Hall mobility ($\mu$) extracted from the Hall curves are shown in Fig. 1f. The $n_s$ is $4.6 \times 10^{13}$ cm$^{-2}$ at 300 K and decreases to a constant value of ~$3.8 \times 10^{13}$ cm$^{-2}$ at low temperatures. This is recognized as a carrier freezing-out effect due to the in-gap states generated by oxygen vacancies[30,47,48]. Meanwhile, the carrier mobility decreases from 140 cm$^2$ V$^{-1}$ s$^{-1}$ at 2 K to 14 cm$^2$ V$^{-1}$ s$^{-1}$ at 250 K due to the lattice vibration, consistent with the previous report on the EuO/KTO (110) heterointerfaces[49].

## Giant enhancement of nonreciprocal transport under ultraviolet irradiation

In order to have a comprehensive understanding of the photoresponse of the CZO/KTO heterointerface, we initially illuminate the sample with the conventional light source at wavelengths ($\lambda$) of 300–700 nm at 5 K. The Hall-bar device is totally covered by the light, as schematically illustrated in Fig. 2a. Prior to systematic light-controllable experiments, the light-induced and large-current-induced heating effects are both ruled out and amorphous-LaMnO$_3$ (a-LMO)/KTO interface maintains insulating during the light illumination (Supplementary Fig. 5), ensuring the intrinsic nature of 2DEGs during light-irradiation transport measurements. The resistance at 5 K firstly decreases by ~7% upon 700-nm illumination, and then shows no significant change in the wavelength range of 700–450 nm due to the low photon energy (Fig. 2b), which largely determines the average location (~2.3 eV) of the existing in-gap state (Fig. 2c). Furthermore, a notable reduction of $R_s$ is observed upon the exerted photon energy between 2.82 and 3.45 eV (i.e., 360 nm ≤ $\lambda$ ≤ 440 nm), which largely determines the average location (~3.1 eV) of the other in-gap state (Fig. 2c). The electrons

residing in these two in-gap states generated from oxygen vacancies can be both excited to the conduction band minimum (CBM), thus increasing the conductivity[50,51]. Particularly, as the wavelength continues to reduce, $R_s$ undergoes a sharp decrease and reaches a minimum at 330 nm (Fig. 2b). This photon energy of ~3.76 eV is just slightly greater than the KTO's bandgap energy (~3.6 eV), thus capable of pumping abundant electrons from the valence band maximum (VBM) to the CBM. The final increase in $R_s$ upon 300-nm illumination is noteworthy. In this case, the electrons in the VBM are excited to the higher subbands in the conduction band due to the larger photon energy (~4.14 eV). These photocarriers lead to increased interactions between electrons and the lattice (phonons), which results in a longer time for the electrons to return to a lower energy state. During this enhanced relaxation process, more photogenerated electrons would experience a notable competition between the weak localization and WAL upon the 300-nm excitation (see the next section) in addition to the likely recombination of photogenerated electrons and holes, thus resulting in a decrease in the total carrier density responsible for conductivity and ultimately an increase in $R_s$.

Since the CZO/KTO shows the distinct photoresponse at different wavelengths, we select light with specific wavelengths to control the nonreciprocal transport. The measurements are carried out with an exerted small direct current ($I = \pm40$ μA) at 5 K with simultaneously applying an in-plane magnetic field normal to the current direction. The current-direction-dependent in-plane MR under illumination is displayed in Fig. 2d. The heterostructure shows a relatively weak nonreciprocal response at $\lambda \geq 500$ nm and in the dark, implying an indistinctive excitation at long wavelengths. This is coincident with the case revealed by the $\lambda$-dependent $R_s$ measurements (Fig. 2b). With the decreasing $\lambda$, a distinct noncoincidence of MR emerges. Remarkably, when illuminated at 330 nm, a significant misalignment of MR is clearly seen upon reversing the current direction, showing a giant enhancement of nonreciprocal transport. We plot the ratio of the resistance change ($\Delta R/R_0$) as a function of the magnetic field in between $+I_x$ and $-I_x$ (Fig. 2e), where the $\Delta R$ represents the difference in resistance upon opposite directions of the currents and $R_0$ represents the longitudinal resistance at zero field. Compared to the dark case, an amplified signal of unidirectional resistance with the magnitude of around 4% is observed owing to the notable gating effect at 330 nm. Besides, the $\Delta R/R_0$ exhibits a linear dependence on the magnetic field as well as the applied current (Supplementary Fig. 6) near the origin, which is known as bilinear magnetoresistance (BMR). The BMR is fitted well with the above-mentioned phenomenological expression (Eq. 1) and has been widely discovered in other noncentrosymmetric systems[23,24]. Meantime, the $\Delta R/R_0$ undergoes a gradual saturation with increasing field, which is likely attributed to the complex effect of the large Zeeman energy exerting on the band structures[52,53]. Moreover, the nonreciprocal transport gradually weakens with increasing temperature and vanishes at around 40 K, which is attributed to the disruption of spin-momentum locking by quantum fluctuations (Supplementary Fig. 7). In addition, we also perform the second harmonic measurements to further confirm the light-induced giant enhancement of nonreciprocal transport at KTO (111)-based interfaces (Supplementary Fig. 8).

The nonreciprocal coefficient $\gamma = \Delta R/(2BIR_0)$ is derived from the phenomenological Eq. (1), which is used as the main figure of merit to quantify the nonreciprocal transport. Since $\Delta R/R_0$ exhibits an excellent linear dependence in ±0.7 T range, we apply this field range to attain $\gamma$. Its evolution with light wavelength is plotted in Fig. 2f. In the dark, the $\gamma$ is estimated to be ~$10^2$ A$^{-1}$ T$^{-1}$, which is comparable to that in LaAlO$_3$/SrTiO$_3$ heterostructures[22] and much larger than those observed in other asymmetric systems, such as BiTeBr[26], $\alpha$-GeTe[15], and Bi$_2$Se$_3$[54]. Very strikingly, the $\gamma$ increases rapidly to ~$10^5$ A$^{-1}$ T$^{-1}$ under 330-nm irradiation, almost three orders of magnitude larger than that in the dark. This giant $\gamma$ value in 2DEGs is notably comparable to those in complex topological insulator nanowires[53] and noncentrosymmetric

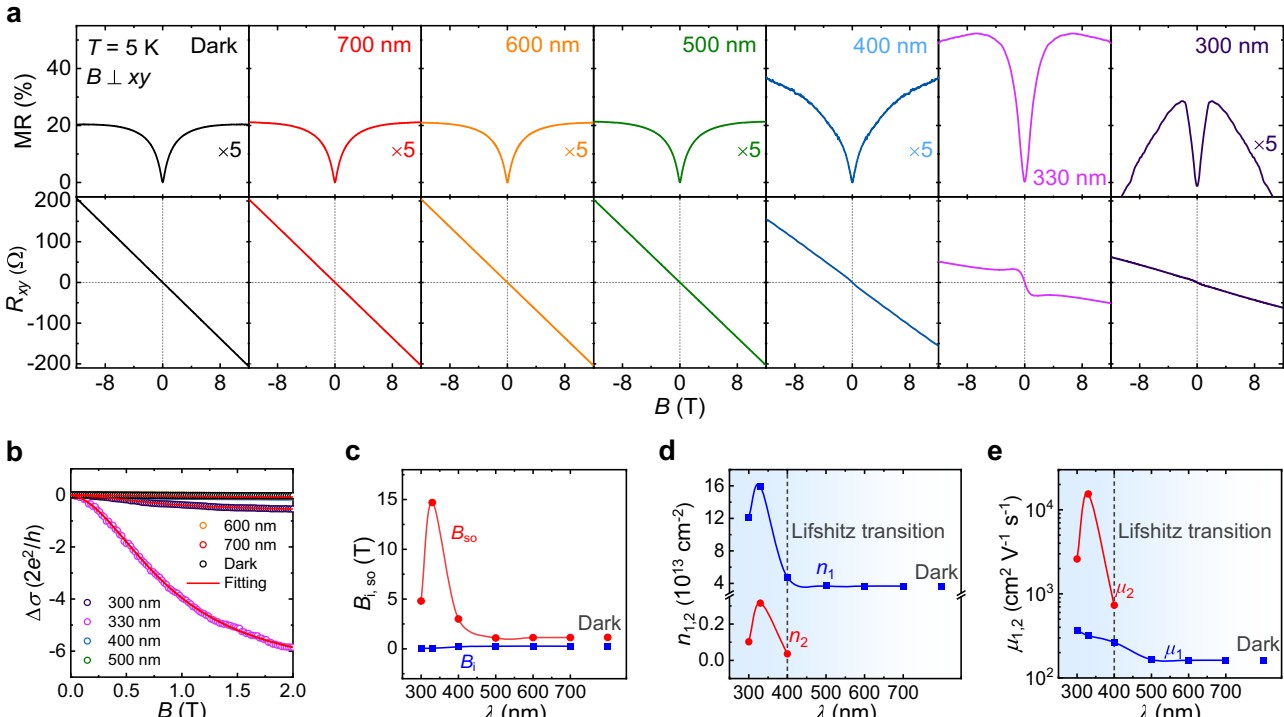

**Fig. 3 | Optical gating of the CZO/KTO heterointerface. a** The MR and Hall curves in the dark and under illumination with various wavelengths under the applied out-of-plane magnetic field at 5 K. **b** The magnetoconductance curves in the dark and at different wavelengths. The solid red curves are the best fits to the MF model. **c** The Rashba SOC effective field ($B_{so}$) and inelastic scattering field ($B_i$) extracted from (**b**) as a function of wavelength. **d, e** The derived carrier density ($n_{1,2}$) and mobility ($\mu_{1,2}$) from the Hall effect as a function of wavelength according to the two-band model. The dashed lines indicate the Lifshitz transition point. The corresponding parameters of the dark condition are also included in (**c**–**e**).

superconductors[55], further establishing the giant tunability of optical control in our nonreciprocal transport. To highlight the giant enhancement of nonreciprocal transport by light control, we summarize the nonreciprocal transport coefficient modulated by various methods, such as electric-field control and ionic-liquid gating in Fig. 2g. It is seen that our light-control method clearly distinguishes itself from this benchmark chart in terms of both the giant $\gamma$ and the extraordinary tunability.

## Light-induced enhancement of Rashba SOC strength and photocarrier excitation

Why does light have such a superior manipulation capacity of non-reciprocal transport? To further reveal the underlying physical picture, we carried out MR and Hall-effect measurements by applying an out-of-plane magnetic field with the identical wavelengths. As shown in Fig. 3a, MR and $R_{xy}$ maintain constant at $\lambda = 700$, 600, and 500 nm as compared to the dark condition, consistent with the above results of $\lambda$-dependent $R_s$ and nonreciprocal transport (Fig. 2b, d). Under 400-nm illumination, MR shows a modest increase and the Hall effect experiences a linear to nonlinear transition due to the excitation of electrons from in-gap states. Remarkably, a much stronger WAL effect emerges at 330 nm, implying a giant Rashba SOC strength. Besides, Hall curve becomes apparently nonlinear. Two distinct slopes can be traced at low and high magnetic fields, respectively, which is a featured behavior of Hall effect contributed by two types of carriers[56]. It is noteworthy that the WAL effect and nonlinear Hall characteristic are simultaneously suppressed when the wavelength decreases to 300 nm, and the competition between the weak localization and WAL becomes notable, which is due to the same physical mechanism as the $R_s$ increase (Fig. 2b). We do not observe any hysteresis in the MR and Hall curves under 330-nm irradiation (Supplementary Fig. 9), indicating that light irradiation cannot induce any spin polarization in KTO-based

2DEGs. It is apparent that there is a close connection among the light-induced MR change, Hall effect, and nonreciprocal transport.

To quantify the Rashba SOC tuned by different wavelengths, the WAL effect is analyzed by employing the modified Maekawa-Fukuyama (MF) theory[57,58], which has been proved to be well applicable in 2DEGs at oxide heterointerfaces[59]. We present the magnetoconductance $\Delta\sigma(B) = \sigma(B) - \sigma(0)$ in the unit of the quantum conductance ($G_Q = 2e^2/h$) under different wavelengths, where $\sigma(0)$ is the magnetoconductance at zero field (Fig. 3b). The total magnetoconductance under the negligible Zeeman splitting can be written as[57,58]

$$\frac{\Delta\sigma(B)}{G_Q} = -\frac{1}{2}\Psi\left(\frac{1}{2}+\frac{B_i}{B}\right)+\Psi\left(\frac{1}{2}+\frac{B_i+B_{so}}{B}\right)-\ln\left(\frac{B_i+B_{so}}{B}\right)$$
$$+\frac{1}{2}\Psi\left(\frac{1}{2}+\frac{B_i+2B_{so}}{B}\right)-\frac{1}{2}\ln\left(\frac{B_i+2B_{so}}{B_i}\right)-\frac{AB^2}{1+CB^2} \quad (2)$$

where $\Psi(x)$ is the digamma function, and $B_i$ and $B_{so}$ are the inelastic scattering and the SOC effective fields, respectively. These two characteristic fields are introduced to characterize $B$-dependent quantum correction based on the inelastic scattering and SOC scattering processes. The last term is from the Kohler's rule, which describes the ordinary magnetoconductance with fitting parameters of $A$ and $C$. The fitting $B_{so,i}$ results are displayed in Fig. 3c. Obviously, $B_{so}$ reaches a maximum of 14.7 T at 330 nm, suggesting a greatly enhanced Rashba SOC strength. Note that the $\lambda$-dependent $B_{so}$ evolution is consistent with that of the $\lambda$-dependent $R_s$ (Fig. 2b), indicating their inherent direct relation and the same physical mechanism. In contrast, the inelastic scattering field ($B_i$) displays a gradual increase with increasing wavelength and is much smaller than the SOC field when $\lambda \le 400$ nm (Fig. 3c), implying the predominant role of the WAL effect. Other deduced parameters (i.e., spin relaxation length, dephasing length, spin relaxation time, inelastic scattering time, Rashba coefficient and

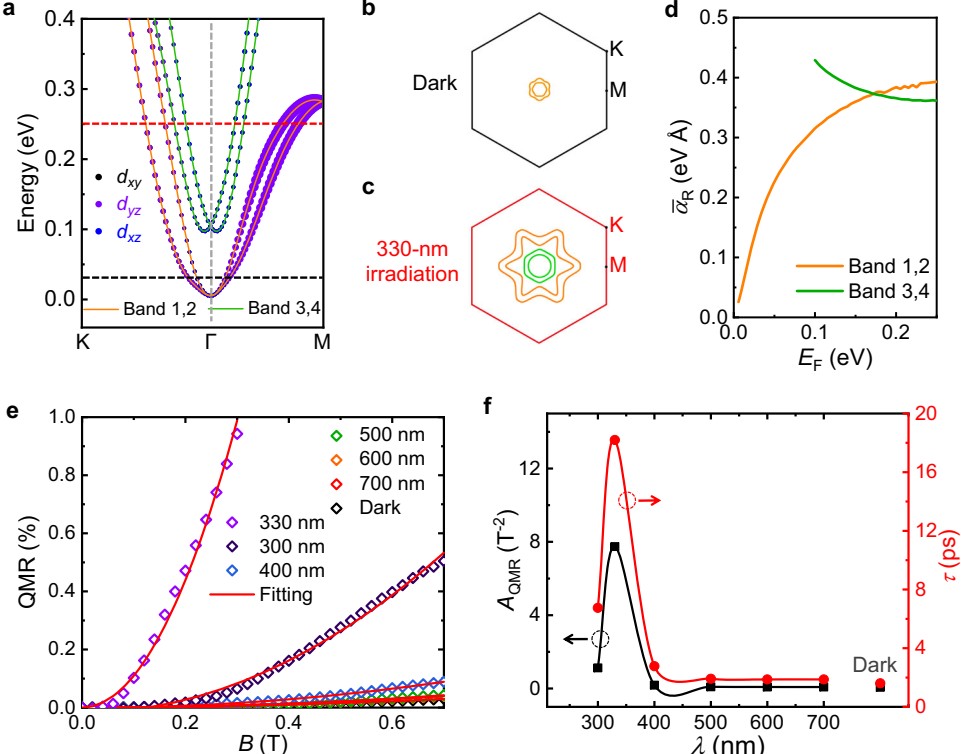

**Fig. 4 | Theoretical analyses for the light-induced enhancement of non-reciprocal transport. a** The calculated electronic band structure (Band 1-4, orange and green lines) and the weight of $d_{xy}$, $d_{yz}$ and $d_{xz}$ (black, purple and blue circles) orbitals of 12-Ta layers of KTO (111) surface from first-principles calculations with Hubbard $U = 4$ eV. The weight of the orbitals is expressed by the size of the corresponding circles. **b, c** The calculated Fermi surfaces at $E_F = 0.03$ (in the dark) and 0.25 eV (with the 330-nm irradiation), which are indicated by the dashed black and red lines in (**a**), respectively. **d** The average Rashba coefficient ($\bar{\alpha}_R$) versus the $E_F$ for bottom Band 1,2 (orange line) and top Band 3,4 (green line). **e** The QMR as a function of the in-plane magnetic field at various wavelengths and in the dark. The solid red curves are the best fits according to Eq. (5). **f** $A_{QMR}$ and the relaxation time ($\tau$) deduced from (**e**) as a function of wavelength. The corresponding parameters of the dark condition are also shown.

spin-splitting energy) are provided in Supplementary Fig. 10 to shed more light on the light-tunable Rashba SOC. When calculating the Rashba coefficient and spin-splitting energy, the effective mass $m^*$ is set to $0.3\,m_0$ (where $m_0$ is the free electron mass) according to the previous angle-resolved photoemission spectroscopy (ARPES) experimental result[60]. It should be noted that the Rashba coefficient ($\alpha_R$) reaches the maximum value of 0.26 eV Å, largely comparable with that deduced from the ARPES experiment[38]. The largest spin-splitting energy ($\Delta$) can also reach 167.9 meV, which is a typically high record among Rashba interfacial systems. Such a highly tunable capacity by light manifests the potential applications of KTO-based interfaces in the energy-efficient opto-spintronic devices.

Furthermore, the two-band model is performed to extract the carrier density and mobility from Hall measurements to figure out the electron excitation process[30,35]:

$$R_{xy}(B) = -\frac{1}{e}\frac{\left(\frac{n_1\mu_1^2}{1+\mu_1^2 B^2} + \frac{n_2\mu_2^2}{1+\mu_2^2 B^2}\right)B}{\left(\frac{n_1\mu_1}{1+\mu_1^2 B^2} + \frac{n_2\mu_2}{1+\mu_2^2 B^2}\right)^2 + \left(\frac{n_1\mu_1^2}{1+\mu_1^2 B^2} + \frac{n_2\mu_2^2}{1+\mu_2^2 B^2}\right)^2 B^2} \quad (3)$$

with the constraint of

$$R_0 = \frac{1}{e(n_1\mu_1 + n_2\mu_2)} \quad (4)$$

where $n_{1,2}$ denote the first- and second-type sheet carriers at the CZO/KTO interface, respectively, $\mu_{1,2}$ are the corresponding Hall mobilities, and $e$ is electron charge. The fitting curves are depicted in

Supplementary Fig. 11 and the fitting parameters are presented in Fig. 3d, e. In the region of $\lambda > 400$ nm, the intrinsic electrons ($n_1$) dominate exclusively for the conductivity with a mobility of 162 cm² V⁻¹ s⁻¹. Nevertheless, a second type of carriers begins to emerge when $\lambda \leq 400$ nm. Remarkably, the carrier density ($n_2$) and mobility ($\mu_2$) of the second-type carriers both undergo a giant increase at 330 nm to $0.3 \times 10^{13}$ cm⁻² and 15400 cm² V⁻¹ s⁻¹, respectively, which stems from the photoelectrons excited from the KTO's VBM under the significant light-excitation process (Fig. 2c). The Lifshitz transition is thus believed to occur at about 400-nm irradiation (Fig. 3d, e). It is noted that the carrier density of $n_1$ undergoes an unexpected increase to $1.6 \times 10^{14}$ cm⁻² at 330 nm, which is very different from the previous observation on the optical gating of oxide 2DEGs where the intrinsic carrier $n_1$ keeps almost constant under illumination[35]. This phenomenon is attributed to the fact that the photocarriers tend to fill the Ta-$t_{2g}$ CBM where the intrinsic carrier occupies, thus increasing $n_1$ remarkably (Supplementary Fig. 12). The sudden decrease in carrier density ($n_{1,2}$) and mobility ($\mu_2$) with 300-nm excitation is also ascribed to the notable competition between the weak localization and WAL as well as the likely recombination of photogenerated electrons and holes, in good agreement with the $R_s$ increase (Fig. 2b). This gives us a comprehensive and in-depth understanding of the complex interplay between light and electron excitation process at the CZO/KTO heterointerfaces, which helps to figure out the origin behind the giant $\gamma$ enhancement.

## Theoretical analyses

In order to disclose the nature of such a giant enhancement of non-reciprocal transport, we perform the density functional theoretical

(DFT) calculations on 2DEGs occurring at the surface of KTO (111) (see "Methods" section for details). The prominent Rashba spin splitting is observed near $\Gamma$ point in the calculated electronic band structure (Fig. 4a). Note that there is no significant difference among calculated electronic band structures of KTO under different Hubbard $U$ values (3–5 eV), suggesting that light only impacts on the location of $E_F$ (Supplementary Fig. 13), thereby influencing the Rashba SOC strength as well as the nonreciprocal transport. Representative Fermi surfaces with $E_F = 0.03$ and 0.25 eV are displayed in Fig. 4b, c, which are marked with the dashed black and red lines (Fig. 4a), respectively. These two Fermi surfaces denote the dark and 330-nm illumination conditions, respectively. In the dark condition, the electronic band filling is low ($E_F = 0.03$ eV). Thus, a Fermi contour is obtained where the Rashba spin-splitting energy ($\Delta$) is relatively small with $\Delta = 35$ meV (Fig. 4b). Upon illumination with 330-nm light, the significant photoexcitation process brings about a sharp increase in band filling ($E_F = 0.25$ eV). In this regard, the Fermi level crosses the bottom of the upper subband, and a complicated pattern of Fermi contour is obtained with the larger $\Delta$ ($\Delta_1 = 310.9$ meV for Band 1,2 and $\Delta_2 = 125.9$ meV for Band 3, 4) (Fig. 4c). As depicted in Fig. 4d, the calculated average $\bar{\alpha}_R$ exhibits a strong dependence on the $E_F$ for Band 1,2. When subjected to illumination at 330 nm, abundant photocarriers start to fill the additional Ta-$t_{2g}$ conduction band (Supplementary Fig. 12). Thus, the $\bar{\alpha}_R$ contributed by all the bands increases nearly one order of magnitude as compared to that in the dark (Fig. 4d), which should contribute to the larger $\gamma$ in view of the criterion of $\gamma \propto \alpha_R \tau^2 / E_F$ [23]. Here, the nonreciprocal transport coefficient $\gamma$ is derived from the underlying physical picture of the nonreciprocal transport[23] rather than the direct phenomenological Eq. (1)[9,10].

Furthermore, the second type of photocarriers has the remarkably high mobility (Fig. 3e), which is mainly excited to Band 3, 4 of the Ta 5d $t_{2g}$ orbitals ($d_{xy}$, $d_{xz}$ and $d_{yz}$) that are highly degenerate. Its high mobility leads to the longer relaxation time $\tau$, which also contributes to the giant $\gamma$ enhancement. Given that the magnitude of $\tau$ is not apt to be estimated from mobility with two types of carriers, we adopt the quadratic magnetoresistance (QMR) to fit it (Fig. 4e). The QMR is defined as $\left[MR(B, +I_x) + MR(B, -I_x)\right]/2$, and it is fitted by the following expression[23]:

$$QMR = \frac{3}{4}\left(\frac{g\mu_B}{\hbar}\right)^2 \tau^2 B^2 \qquad (5)$$

where $\frac{3}{4}\left(\frac{g\mu_B}{\hbar}\right)^2 \tau^2$ is defined as $A_{QMR}$, $g$ is the g-factor, $\mu_B$ is the Bohr magneton and $\hbar$ is the reduced Plank constant. Since the $A_{QMR}$ is independent of $\alpha_R$ and $E_F$, the $\tau$ can be deduced from $A_{QMR}$. It is seen that the $\tau$ at 330 nm increases ten times than that in the dark (Fig. 4f). Consequently, it is a synergistic effect resulting from both the light-induced increased Rashba SOC strength and the prolonged relaxation time that eventually boosts the substantial enhancement of nonreciprocal charge transport at KTO (111)-based interfaces.

In summary, we have achieved a giant enhancement of nonreciprocal charge transport via optical gating at the Rashba heterointerfaces of CZO/KTO. The conventional light source with a specific wavelength of 330 nm pumps abundant electrons from the valence band of KTO to the Ta-$t_{2g}$ conduction band, thus bringing about the stronger Rashba SOC strength as well as additional high-mobility photocarriers. Thanks to the above remarkable photodoping effect, the nonreciprocal transport coefficient $\gamma$ experiences three orders of magnitude increase up to ~$10^5$ $A^{-1} T^{-1}$, which is well confirmed by DFT theoretical calculations. Our work not only provides an alternative approach to control the nonreciprocal transport by the external light stimulus in various Rashba systems but also overcomes the bottleneck of previous methods to facilitate the potential applications in opto-rectification devices and spin-orbitronic devices.

## Methods

### Film fabrication
The films were fabricated on $5 \times 5$ mm² (111) KTO single-crystal substrates by the PLD technique using 248-nm KrF laser with a fluence of 2 J cm⁻² and the repetition rate of 1 Hz. Before film growth, the substrates were cleaned with acetone, alcohol, and deionized water successively. The Hall-bar device was fabricated by optical lithography using a hard mask made from a-LMO with width of 50 µm and length of 500 µm. The 50-nm-thick CZO films were epitaxially deposited at 750 °C with the oxygen pressure of $10^{-5}$ mbar. After deposition, the films were cooled down to room temperature at 5 °C per minute under the same oxygen pressure.

### Structural characterization
The crystal structure of the films was measured by $\theta-2\theta$ XRD in a high-resolution mode using Cu-$K_\alpha$ emission (1.5418 Å, Bruker D8 Discover). The cross-sectional specimen for STEM-HAADF was prepared on a ZEISS Crossbeam 550 L electron beam/focused ion beam electron microscopy at 1–30 kV. The high-resolution STEM images were collected by the aberration-corrected microscope operated at 200 kV (JEOL ARM 200 F). For STEM-HAADF imaging, the semi-convergent angle of the probe-forming lens was set to 25 mrad. The EDX mapping was acquired from 2 silicon drift detector system.

### Light-irradiation transport measurements
Prior to the light-irradiation transport measurements, electrodes were bonded with aluminum wires using an ultrasonic wire bonder. The standard eight-probe configuration (Fig. 2a) was used for the transport characterization through a 12 T Cryogenic cryogen-free measurement system (CFMS-12). Keithley 6221 and 2182 nanovoltmeters were employed for current source and voltage source detection, respectively. To study the effect of light control during the transport measurements, we used a tunable conventional xenon light source with the wavelength ranging from 300 to 700 nm. The light beam was guided into the CFMS transport system through the sample holder rod attached with a quartz optical fiber. When the heterostructure was exposed to the light, the light absorption in the upper CZO layer was negligible due to its wide-bandgap property and transparent nature. The light power density was maintained a constant value of ~0.3 mW cm⁻² for all the wavelengths to avoid any light-induced heating effect. Notably, after each circle of the light irradiation with a certain wavelength, CZO/KTO heterostructures were warmed up to the room temperature to restore the initial state. Thus, the persistent photoconductivity could be ruled out. In addition, we performed the light-controlled experiments after waiting for enough time to stabilize the resistance every time. As the nonreciprocal transport is more pronounced at low temperatures, the temperature for the transport measurements was mainly set to 5 K.

### First-principles calculations
We performed the DFT first-principles calculations upon electronic properties of KTO of slab models by employing the Vienna ab initio simulation package (VASP)[61,62]. A slab model of 12 Ta-layers on the (111) surface was considered. We adopted the generalized gradient approximation (GGA) with the Perdew-Burke-Ernzerhof[63,64] type exchange-correlation potential with the energy cutoff fixed to 450 eV. By considering the transition metal Ta, GGA plus Hubbard $U$ functions with $U = 3$, 4, and 5 eV were applied for Ta-$d$ orbitals for all the results. We supplied the atomic coordinates of the optimized computational models for $U = 3$, 4, and 5 eV in Supplementary Data 1. Total Hellmann-Feynman forces were taken to $10^{-2}$ eV A⁻¹ for the structural optimization. $6 \times 6 \times 1$ $\Gamma$-centered $k$-points sampling was used in the primitive unit cells. A total energy tolerance $10^{-7}$ eV was adopted for self-consistent convergence. The SOC was included in self-consistent calculations and energy band calculations. In order to obtain the Fermi

surface, we used the maximally localized Wannier functions (MLWF) from the first-principles calculations[65,66] to set up tight-binding Hamiltonians. The Ta-$d$, O-$p$ orbitals were initialized for MLWFs by Wannier90[67]. We calculated the average Rashba coefficients from the energy spectrum

$$\bar{\alpha}_R = \frac{\hbar^2 \Delta \bar{k}}{2m^*} \tag{6}$$

where $\Delta \bar{k} = \bar{k}_{in} - \bar{k}_{out}$ is the difference of two neighboring subbands, and $m^*$ is the electron effective mass.

## Data availability
All data that support the key findings in this study are available within the main text and the Supplementary Information file. Additional raw data are available from the corresponding authors upon request. Source data are provided with this paper.

## Code availability
The code that supports the theoretical plots within this paper is available from the corresponding authors upon request.

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

## Acknowledgements

This work was supported by the National Key Research and Development Program of China (grant no. 2022YFA1402404 to X.W.) and the National Natural Science Foundation of China (grant nos. T2394473, T2394470, 62274085, 11874203 and 61822403 to X.W., grant nos. 92161201 and 12025404 to F.S., and grant nos. 92265203 and 11974340 to W.L.).

## Author contributions

X.W. conceived the study and proposed the strategy. X.W. and R.Z. supervised the project. X.Z. and Z.C. grew the samples and performed XRD measurements. T.Z. and H.Z. conducted theoretical calculations. Y.T. and B.G. carried out the microscopic characterization. X.Z., A.S., C.Z. and R.G. fabricated the devices and performed transport measurements. G.L. and J.S. performed superconductivity measurements. S.Z., W.N., Y.C., F.F., W.L., K.C., F.S. and R.Z. contributed to the data analysis and discussion. X.W. and X.Z. wrote the manuscript with input from all the authors. All authors contributed to manuscript revisions.

## Competing interests

The authors declare no competing interests.
