## [Peer Review File · Nature Communications]

Light-induced giant enhancement of nonreciprocal transport at KTaO₃-based interfacesREVIEWER COMMENTS

Reviewer #1 (Remarks to the Author):

The authors report the observation of a giant enhancement of nonreciprocal transport in the Rashba 2D electron gases at the CaZrO₃/KTaO₃ (111) interfaces: the coefficient is as high as $10^5 \text{A}^{-1} \text{T}^{-1}$. The study on the nonreciprocal transport has great significance for both fundamental physics research and potential device application. However, there are unconvincing conclusions and explanations listed below.

(1) It is well known that the resistance of the 2DEG upon the light illumination might slowly decay with time, which would greatly affect the results of time-consuming measurements (for instance, magnetoresistance, Hall data). Besides, the light or large current induced heating also has the accumulation effect, which is another factor that would impact the accuracy of the results. It is necessary to demonstrate that the resistance is steady enough during the illumination or upon large current stimulation.

(2) In Fig. 3a, the Hall resistance with the 330 nm light illumination seems not like the two-carrier transport behavior (generally featured with the "S" shape), but more complicated (more kinks). Thus the two-carrier model fitting is not convincing. Moreover, the Ta 5d t_{2g} orbitals (d_{xy}, d_{xz}, d_{yz}) are highly degenerate for the (111) orientation, while the author did not elucidate the origin of the second type of carriers. Further, the derived two types of carriers show a great discrepancy (two orders of magnitude) in the mobility, which is also contradictory to the DFT results in this work (no signs for vastly different light/heavy bands).

(3) The derived H_{SO} for the 330 nm illumination (15T) is doubtful since the fitting range is too narrow (0 - 0.5T), considering the WAL fitting is quite sophisticated.

(4) The sudden drop of for the carrier density, H_{so} for the 300 nm illumination is also not reasonable if the photon flux is kept constant for varied wavelength. Otherwise, if the photon flux is not kept constant, the comparison for different wavelength is not appropriate.

(5) The giant enhancement of the nonreciprocal transport might also origin from the inhomogeneity of the carrier distribution at the interface. That is, the authors should provide sufficient proof that the conducting channel is homogeneous and the LMO/KTO interface (the KTO surface underneath the hard mask) always maintains insulating during the light illumination.

(6) One suggestion: the authors can further verify the nonreciprocal transport behavior with the conventional second harmonic measurement.

Thus, the manuscript should address the above questions for publication in Nature Communications.

Reviewer #2 (Remarks to the Author):

Please refer to the attached comments.

Reviewer #3 (Remarks to the Author):

By means of light irradiation, Zhang et al. reported a giant enhancement of nonreciprocal transport in the Rashba 2D electron gases at superconducting and epitaxial CaZrO₃/KTaO₃(111) interfaces. The nonreciprocal transport coefficient undergoes a giant increase with three orders of magnitude up to $10^5 \text{ A}^{-1}\text{T}^{-1}$. Furthermore, a strong Rashba spin-orbit coupling (SOC) effective field of 15 T is achieved with abundant high-mobility photocarriers under ultraviolet illumination, which accounts for the giant enhancement of nonreciprocal transport coefficient. The first-principles calculations further disclose the stronger Rashba SOC strength and the longer relaxation time in the photoinduced carrier excitation process, bridging the light-property quantitative relationship. Their work provides a new pathway to boost nonreciprocal transport in noncentrosymmetric systems and facilitates the promising applications in rectification devices and spin-orbitronic devices.

The manuscript has grown CaZrO₃/KTaO₃(111) 2D electron gases and achieved a superconducting transition temperature of 0.43 K. In addition, ultraviolet irradiation was used to modulate nonreciprocal transport, achieving a significant enhancement of nonreciprocal transport. Compared with modulation methods of other systems, the CaZrO₃/KTaO₃(111) 2D electron gas in this manuscript has the high nonreciprocal transport coefficient, which is a good innovation. However, the manuscript lacks a certain data fitting process and derived formulas, and the theoretical part is not specific enough. I think more detailed explanations and improvements in details are needed. Without clarifying these key issues, the paper unfortunately did not meet the criterion of Nature Communications at the current stage.

The specific issues are as follows:

1. The first thing I want to know is, is there any magnetic hysteresis in the MR-H and R_{xy}-H curves under light irradiation?
2. Line-105: The manuscript states that there is no elemental interdiffusion in Figure 1c, but there is obvious diffusion of Ca in KTO, which is inaccurate.
3. Line-131: The manuscript does not specify whether the light intensity (power) is consistent at different wavelengths.
4. Line-134: When the wavelength of light is less than 330nm, please explain why R_s increases as the wavelength decreases? (Figure 2b) Supplementary material S3 was not cited in the main text, nor did it explain the selection of temperature (measuring physical quantities such as R_s at T=5 K).
5. Figure 2d: Missing unit in y-axis MR(%).
6. Figure 2c: What do the energies of -1.7eV and -3.0eV represent in the Schematic? What is the basis for obtaining these data?
7. Line-198: In this paper, the Hall resistance has two different slopes, which is used to determine the Hall effect behavior contributed by two types of carriers. There is a lack of specific fitting formula, image and fitting curve. In addition, there is no detailed explanation for the changes in carrier concentration n₁ and n₂ and mobility μ_1 and μ_2 obtained after fitting (Fig. 3d,e). For example, when the wavelength is less than 330nm, the reasons for the decrease of n₁, n₂ and μ_2 and the increase of μ_2 are not explained.

8. Figure 3b: Lack of interpretation of figure.
9. Line222: It is written that the second type of carrier appears at 330nm. However, according to Figure 3d, when $330\text{nm} < \lambda < 400\text{nm}$, the concentration of the second type of carrier n_2 is not zero, indicating that the second type of carrier has appeared when the wavelength is just less than 400nm.
10. Line-224 and Figs.3d and 3e: How are Lifshitz transition points selected? (Figure 3c, d, e) What is the theoretical basis for the Lifshitz transition points? Or are these just speculation? Are they related to the electronic band structure of KTO(111) in any way? Please clarify.
11. Line-259: The image reference is incorrect, it should be Figure 4e instead of Figure 4d.
12. Figure 4a: Compared with dark conditions, the electronic band structure will change under light conditions[see, e.g., Ref.27]. This manuscript did not calculate the electronic band structure under two different conditions, nor did it indicate whether Figure 4a is the band structure in the dark or under light conditions. It is recommended to supplement the electronic band structure and the weights of d_{xy} , d_{xz} , and d_{yz} under the two conditions.
13. Figure 4e: What do the orange and green curves represent respectively? Please add which corresponds to AQMR and which corresponds to relaxation time τ ?

In "Light-induced giant enhancement of nonreciprocal transport at KTaO₃-based interfaces," the authors used light irradiation to observe the enhancement of nonreciprocal transport in the Rashba SOC in 2DEG at the interface of superconducting and epitaxial CaZrO₃/KTaO₃ interfaces. With ultraviolet illumination, the authors observe an enhancement in the Rashba SOC effective field due to the larger Rashba coefficient and longer carrier relaxation time. The authors demonstrated the interplay of light and nonreciprocal transport. Transport properties were characterized through Hall bar measurements. The authors measured the WAL pattern and fitted it with the Maekawa-Fukuyama (MF) model to extract the Rashba SOC strength through measured WAL patterns. This work is well structured and presents the significant experimental results clearly. However, there are quite a few major concerns to be addressed before it can be considered further for Nature Communications. Please see comments below:

Questions and comments:

Major concerns:

1. The authors only use a rather short passage (lines 70-78) on page 4 to introduce 2DEGs in a KTO-based interface. In my opinion, the authors should introduce more 2DEG properties in this interface and its connection to this work in a more in-depth manner. By doing so, readers who are unfamiliar with the KTO-based interface can better grasp the essence of this material and its applications.
2. In lines 202-204 on page 8, the authors claim that there are close connections between the magnetoresistance (WAL effect), Hall effect, and nonreciprocal transport. Although there are relevant discussions in the following sections and some discussions in lines 266-271, it is quite difficult to grasp this "connection" and their underlying physics. For instance, in Figs. 2 and 3, the authors did point out that when the illuminated light varies from 400 nm to 300 nm, the WAL is enhanced significantly, but is then quickly suppressed. In the meantime, when a 330 nm light incident on the Hall bar, a nonlinear Hall resistance emerges due to the existence of two different carrier types. In addition, from Fig. 3 and 4, the authors suggest that there is a Lifshitz transition point in nonreciprocal transport coefficient, carrier density, and mobility. Are these three effects/observations connected or related in some way? Can they all be explained using the Ta-t_{2g} conduction band model mentioned in lines 251 and 268? The reviewer recommend that the authors should elaborate, or even present figures, to demonstrate the Ta-t_{2g} conduction band in the band structure and how the 330 nm optical pumping leads to electrons jumping from the valance band to the Ta-t_{2g} conduction band.
3. In lines 227-229 on page 9, the authors present a hypothesis to explain the unexpected rise in the density when the incident wavelength is below 400 nm. Is this hypothesis based on any physical models or theories in the existing literature? The explanation isn't clear. For example, what is a two-band? A higher subband in the

quantum well? Second, under light illumination, photo-induced carriers do not necessarily fill the ground state. The ratio of carriers in the excited and ground state would be determined by the pump (illumination) rate and carrier lifetime of transition from the excited state to the ground state. The authors shall discuss with more rigorous calculation to support their great discovery.

4. The explanation of the nonlinear Hall and WAL effect using light with wavelength shorter than 330 nm is not satisfactory. If the authors think bilayer (two-types of carriers) are a major concern for wavelength of 330 nm, why would it lead to the same (similar) results for 300 nm?
5. In line 152, the author applied in-plane magnetic fields, can they explain why? In WAL experiment, a perpendicular (to the 2DEG plane) magnetic field is applied. An in-plane would suppress the WAL effect (see F. E. Meijer et al., Phys. Rev. B, 70, 201307, 2004 or Phys. Rev. Lett., 94, 186805, 2005). Please the authors elaborate more the reasons behind their experiment concept.
6. While γ is defined in p. 4 line 171, but it is not clear why γ is also proportional to another term defined in line 248 (p. 10). A proper explanation with a prior work is necessary.
7. Fig. 1:
 - (a): please label the atomic constituents of the CZO/KTO heterostructure. Also label the orientations.
 - (b): It is difficult to see O atoms in the STEM image. The question is why the authors O at those specific locations? In one unit cell, there should be 3 x O atoms (or multiples). It seems non-sense to only label only 1 O atom in (b).
 - (d): The authors did not provide fabrication details of Hall bar devices. Second question, if a metal gate is deposited, how would light penetrate to the underlying heterostructure. Even though ultraviolet light might not be absorbed (or partially penetrate), the authors should provide the relevant argument and prior work. Third: the transition temperature for superconductivity is not abrupt. It is misleading to claim T_c is 0.43 K.
8. Fig. 2:
 - b: Please explain why there exists a hump of sheet resistance at wavelength of 350 ~ 450 nm? Then why under illumination with an even short wavelength, the sheet resistance is increased?
 - c: Please cite prior work on the in-gap defect states. Furthermore, it is better to show carriers can be excited under light illumination with its photon energy smaller than the bandgap of KTO. For example, carriers might be excited in the valence band at CZO side and jump to the minimum of the quantum well in the KTO. The required energy is smaller than the bandgap of KTO. Please comment on this.
9. Fig. 3(d): The two-band model seems unreasonable. If I look at the values of n_1 and n_2 , it is obvious the contributions from n_1 to the total density is dominant. This will

make the total (effective) mobility close to μ_1 . However, Fig. 1f says another story. The mobility at low temperature is ~ 30 , which is way below μ_1 ($100 \sim 200$) at dark. Please explain.

10. Prior work not cited properly.

- The authors mentioned the importance of their great results on spin-induced SOC effect for spin-orbitronics and rectification device applications, but did not address the potential how their results can be applied to practical device applications. The authors did not cite relevant reference on so-called rectification devices, which hampers readers to further recognize the impact of this work.
- In lin 61 ~ 62 (p. 3), the authors claimed that the gate bias is limited to modulate the SOC strength? Please briefly explain and cite a prior work.
- The citation for the expression in equation (2) is missing. If the authors derived it, please provide the details in the supplementary part.

Minor comments:

11. In line 253 on page 10, the authors mentioned Fig. 2h, but it is missing. Did the authors mean Fig. 3e?
12. Please briefly introduce how Rashba SOC effects contribute to non-reciprocal transport.
13. The authors did not provide the light power or intensity, which is important for the quantitative analysis.
14. In line 105 (p. 5), the EDX data only suggest there exists a hetero-interface, but not a good indicator for the abruptness. I would recommend the authors do not use "abrupt". Instead, "clear" is probably a suitable term to describe their data.
15. In line 112 (p. 5), the transition of resistance does NOT drop to zero abruptly, so using "sharply" isn't appropriate.
16. While the terms of inelastic scattering (H_i) and SOC fields (H_{so}) are defined in equation (2), it is better to briefly introduce the physical meaning of those two terms.
17. For the equation (2), G_0 should be corrected to $2e^2/h$ or $e^2/(\pi \times \hbar)$. r and Δ_{ρ} needs to be clearly defined, too.
18. What is R_0 ? (Line 158). Please define.
19. Discussion must be corrected to "conclusion" since there is NO discussion at all.

RESPONSE TO REVIEWERS' COMMENTS (NCOMMS-23-60568)

We sincerely appreciate the thorough evaluation of our work by all 3 reviewers and their time. Below, we provide the point-by-point response to each comment marked in *blue*. The significant text changes are highlighted in *red* in the revised manuscript.

Report of Reviewer #1 (Remarks to the Author):

The authors report the observation of a giant enhancement of nonreciprocal transport in the Rashba 2D electron gases at the CaZrO₃/KTaO₃ (111) interfaces: the coefficient is as high as $10^5 A^{-1}T^{-1}$. The study on the nonreciprocal transport has great significance for both fundamental physics research and potential device application. However, there are unconvincing conclusions and explanations listed below.

Reply: We are very grateful to Reviewer #1 for the valuable inputs on our work. We are encouraged by the reviewer's remarks on "*the great significance*" of our study "*for both fundamental physics research and potential device application*". In the below, we address the reviewer's thoughtful comments in a point-by-point manner. We hope that the conclusions and explanations have become sufficiently convincing now.

(1) *It is well known that the resistance of the 2DEG upon the light illumination might slowly decay with time, which would greatly affect the results of time-consuming measurements (for instance, magnetoresistance, Hall data). Besides, the light or large current induced heating also has the accumulation effect, which is another factor that would impact the accuracy of the results. It is necessary to demonstrate that the resistance is steady enough during the illumination or upon large current stimulation.*

Reply: Thank you for these meaningful questions. Indeed, the sheet resistance of 2DEGs at the CZO/KTO interface rapidly decreases from 544 to $\sim 37 \Omega/\square$ upon light ON (330 nm) and maintains an almost constant value ($\sim 32 \Omega/\square$) at 5 K with the time evolution (**Fig. R1a**). Thus, our transport results would not be influenced by the resistance decay.

For the light-induced heating that may impact on the accuracy of the results, we believe that this factor can be safely ruled out. Under the same light power (with the power density set at $\sim 0.3 \text{ mW cm}^{-2}$), wavelength-dependent experiment indicates that the excitation of CZO/KTO is selective for the wavelength of the light, and in particular, the giant enhancement of the nonreciprocal transport coefficient observed at 330 nm coincides with the bandgap of the KTO (**Fig. 2**). Furthermore, it is seen from **Fig. R1b** that the ratio of the resistance change ($\Delta R/R_{t=0}$) under 330-nm illumination at 5 K is almost unchanged for at least 2 h, which rules out the light-induced heating effect.

When applying the large current of 100 μA (significantly larger than 40 μA applied in our nonreciprocal transport), the $\Delta R/R_{t=0}$ still keeps largely unchanged at 5 K for at least 2 h regardless of whether the light is OFF or ON (**Fig. R1c**), thus ruling out the large-current-induced and light-induced heating effects. Furthermore, the a-LMO/KTO

interface keeps highly insulating ($\sim 10^6 \Omega/\square$) during the light illumination at 5 K (**Fig. R1d**). The stability tests at 5 K guarantee the reliability and repeatability of the acquired MR, Hall data and nonreciprocal transport from the 2DEGs upon light irradiation.

Fig. R1 | Stability test of the 2DEGs at the CZO/KTO interfaces. **a**, The time evolution of the sheet resistance under the in-situ light illumination at 330 nm at 5 K. The MR and Hall effect measurements are carried out when the sheet resistance is stabilized under the light illumination. **b**, The time evolution of the ratio of the resistance change ($\Delta R/R_{t=0}$) under 330-nm illumination at 5 K. $R_{t=0}$ represents the initial sheet resistance at $t = 0$. $\Delta R = R_t - R_{t=0}$ represents the resistance change. The applied current is 40 μA . A negligible resistance variation of less than 0.5% is observed with the time evolution for at least 2 h, excluding the light-induced heating effect. **c**, The time evolution of $\Delta R/R_{t=0}$ in the dark and under 330-nm illumination at 5 K, respectively. The applied current is as large as 100 μA . The resistance still maintains nearly unchanged (within 0.6%) even under 330-nm illumination at 5 K for at least 2 h, ruling out any light-induced and large-current-induced heating effects during the light-irradiation measurements. **d**, The time evolution of the resistance at the a-LMO/KTO interface under the light illumination at 330 nm at 5 K.

Action: We have added Fig. R1 in Supplementary Information as new Fig. S5. We have added the description in Methods section: “In addition, we performed the light-controlled experiments after waiting for enough time to stabilize the resistance every

time.” (Line 389 on Page 15). We have added the description: “Prior to systematic light-controllable experiments, the light-induced and large-current-induced heating effects are both ruled out, and amorphous-LaMnO₃ (a-LMO)/KTO interface maintains insulating during the light illumination (Supplementary Fig. 5), ensuring the intrinsic nature of 2DEGs during light-irradiation transport measurements.” (Line 157 on Page 7). In addition, we have also added the parameter of light power density in Methods section: “The light power density was maintained a constant value of $\sim 0.3 \text{ mW cm}^{-2}$ for all the wavelengths to avoid any light-induced heating effect.” (Line 383 on Page 15).

(2) In Fig. 3a, the Hall resistance with the 330 nm light illumination seems not like the two-carrier transport behavior (generally featured with the “S” shape), but more complicated (more kinks). Thus the two-carrier model fitting is not convincing. Moreover, the Ta 5d t_{2g} orbitals (d_{xy} , d_{xz} , d_{yz}) are highly degenerate for the (111) orientation, while the author did not elucidate the origin of the second type of carriers. Further, the derived two types of carriers show a great discrepancy (two orders of magnitude) in the mobility, which is also contradictory to the DFT results in this work (no signs for vastly different light/heavy bands).

Reply: We thank the reviewer for raising these important questions. The Hall resistance at 330 nm is plotted in **Fig. R2a** for convenience. The reviewer is correct that the Hall resistance is a little bit abnormal and complicated since there are kinks at around $\pm 1 \text{ T}$. We state that this kink is originated from the significant excitation of the second type of carriers with extremely high mobility at 330 nm. In order to obtain a clear presentation of the excitation process, we show the Hall resistance at various light powers at 330 nm (**Fig. R2b**). It clearly shows the transition from single type carrier (linear Hall resistance) to two types of carriers (nonlinear Hall resistance) as well as the emergence of kinks with increasing light power. We fit the Hall resistance perfectly using the two-band model (equations (3) and (4) in the main text). In addition, similar Hall resistance with kinks under illumination at KTO-based interfaces have also been observed in the previous work (see *ACS Nano* **13**, 609 (2019)). Thus, it can be still regarded as the two-carrier transport behavior, and the two-carrier model fitting is reliable and convincing.

Fig. R2 | Hall curves perfectly fitted using the two-band model. a, The fitted curves of

the Hall resistance at 330-nm illumination using the two-band model. Note that the kinks occurring at the Hall resistance at around ± 1 T of 330-nm illumination are originated from the significant excitation of the second type of carriers with extremely high mobility at 330 nm. **b**, The Hall curves measured at various light powers normalized with that in **a**, indicating the clear excitation process from single type carrier (linear Hall resistance) to two types of carriers (nonlinear Hall resistance) with increasing light power.

We believe that the second type of carriers stems from the photoelectrons excited from the KTO's valence band maximum upon the 330-nm excitation. And these photocarriers are mainly excited to the band 3,4 (green lines, **Fig. 4a**) of the Ta $5d t_{2g}$ orbitals (d_{xy} , d_{xz} and d_{yz}) that are highly degenerate.

The great discrepancy (two orders of magnitude) in the mobility of the derived two types of carriers (**Fig. 3e**) is ascribed to their different relaxation time. Considering the expression $\mu = e\tau/m^*$, where μ represents the mobility, e is the electron charge, τ is the relaxation time, and m^* represents the effective mass, respectively, the mobility is determined by both m^* and τ . Although there is no significant difference between the m^* for different subbands (from **Fig. 4a** of DFT calculations), the second-type electrons occupying the higher subband are less likely to be scattered by disorder since they are far away from the interface (see *Science* **372**, 721 (2021)). Thus, these electrons have a longer relaxation time and the much higher mobility. The remarkable difference of the relaxation time between two types carriers has been further confirmed by our QMR fitting in **Fig. 4f**. The τ at 330 nm is ten times larger than that in the dark. Through the above analysis, we believe that the great discrepancy in the mobility is not contradictory to our QMR fitting as well as DFT calculations.

Action: We have added Fig. R2 in Supplementary Information as new Fig. S11b,d. We have rewritten the sentence with “Nevertheless, a second type of carriers begins to emerge when $\lambda \leq 400$ nm. Remarkably, the carrier density (n_2) and mobility (μ_2) of the second-type carriers both undergo a giant increase at 330 nm to 0.3×10^{13} cm $^{-2}$ and 15400 cm 2 V $^{-1}$ s $^{-1}$, respectively, which stems from the photoelectrons excited from the KTO's VBM under the significant light-excitation process (Fig. 2c).” (Line 282 on Page 11). In addition, we have added the description with: “Furthermore, the second type of photocarriers has the remarkably high mobility (Fig. 3e), which is mainly excited to Band 3,4 of the Ta $5d t_{2g}$ orbitals (d_{xy} , d_{xz} and d_{yz}) that are highly degenerate.” (Line 325 on Page 13).

(3) *The derived H_{so} for the 330 nm illumination (15 T) is doubtable since the fitting range is too narrow (0 - 0.5 T), considering the WAL fitting is quite sophisticated.*

Reply: Thank you for this insightful comment. We apologize for not showing the full fitting range, which is actually between 0 and 2 T, as shown in **Fig. R3a**. Moreover, we show the more Rashba SOC parameters deduced from the WAL fitting in **Fig. S10** of Supplementary Information. Thus, the fitted B_{so} value of 15 T is reliable.

Fig. R3 | The fitted curves of $\Delta\sigma$ according to the MF model in the dark and at various wavelengths. a, The showcase of the full magnetic field range of 0-2 T. **b**, The enlarged figure of a.

Action: We have updated Fig. 3b using the above Fig. R3a and added Fig. R3b in Supplementary Information as new Fig. S10a. In addition, we have added the discussion: “Other deduced parameters (i.e., spin relaxation length, dephasing length, spin relaxation time, inelastic scattering time, Rashba coefficient and spin-splitting energy) are provided in Supplementary Fig. 10 to shed more light on the light-tunable Rashba SOC. It should be noted that the Rashba coefficient (α_R) reaches the maximum value of 0.29 eV Å, largely comparable with that deduced from angle-resolved photoemission spectroscopy experiment³⁸. The largest spin-splitting energy (Δ) can also reach 163.6 meV, which is a typically high record among Rashba interfacial systems. Such a highly tunable capacity by light manifests the potential applications of KTO-based interfaces in the energy-efficient opto-spintronic devices.” (Line 263 on Page 10).

(4) *The sudden drop of for the carrier density, H_{so} for the 300 nm illumination is also not reasonable if the photon flux is kept constant for varied wavelength. Otherwise, if the photon flux is not kept constant, the comparison for different wavelength is not appropriate.*

Reply: Thank you for your insightful comment. We apply the similar experiment concept as the previous work (*Adv. Mater. Interfaces* **7**, 2000646 (2020)) to **keep the light power density constant for all wavelengths**. In addition, we perform the light-controlled experiments after the resistance is stabilized every time (**Fig. R1a**). Therefore, for the each-wavelength excitation, we provide the strong enough photon flux, namely the photon numbers are much larger than those of excited electrons, ensuring the excitation of all electrons. Thus, it is appropriate to compare the carrier density, mobility, and nonreciprocal transport coefficient with different wavelengths. The sudden drop of the carrier density and B_{so} for the 300-nm illumination is also ascribed to the recombination of photogenerated electrons and holes as well as the

downward of the E_F (see **Fig. R6**), in good agreement with the R_s increase (**Fig. 2b**).

Action: The origin of the R_s increase has been discussed in the main text (Line 173-180 on Page 7), which reads: “The final increase in R_s upon 300-nm illumination is noteworthy. In this case, the electrons in VBM are excited to the higher subbands in the conduction band due to the larger photon energy (~ 4.14 eV). These photocarriers leads to increased interactions between electrons and the lattice (phonons), which leads to a longer time for the electrons to return to a lower energy state. During this process, more photogenerated electrons and holes are likely to recombine in a non-radiative manner, resulting in a decrease in the total carrier density and ultimately an increase in R_s .”

(5) *The giant enhancement of the nonreciprocal transport might also origin from the inhomogeneity of the carrier distribution at the interface. That is, the authors should provide sufficient proof that the conducting channel is homogeneous and the LMO/KTO interface (the KTO surface underneath the hard mask) always maintains insulating during the light illumination.*

Reply: Thank you for your professional comment. We provide the sufficient evidence that the conducting channel is homogeneous from two aspects. **First**, we carry out the resistivity measurements from the arbitrarily selected Hall-bar electrodes. As seen from **Fig. R4a**, there is no significant difference of the resistivity between each electrode in both dark condition and under 330-nm illumination, indicating the excellent homogeneity of the conducting channel of 2DEGs. Besides, no remarkable difference is observed when applying the opposite-direction current, further ruling out the inhomogeneity of the carrier distribution. **Second**, we perform the anisotropic MR with the measurement configuration shown in the inset of **Fig. R4b**. The sharp peaks at 90° and 270° when the field is applied parallel or antiparallel to the current indicate the 2D conduction characteristic and also the uniformity of the 2DEGs at the interface. We can estimate the thickness of the conduction layer (t) by adopting the formula of $t = h/(e^2 k_F R_s \sqrt{\alpha})$, where h is the Planck constant, e is the electron charge, k_F is the Fermi wave vector ($k_F = \sqrt{2\pi n_s}$), R_s is the sheet resistance, and α is defined by the ratio of the perpendicular MR to the parallel MR. The conduction layer thickness of KTO-based 2DEGs is thus estimated to be ~ 4.35 nm.

The a-LMO/KTO interface does maintain highly insulating ($\sim 10^6 \Omega/\square$) during the light illumination, as seen from the time evolution of the resistance at the a-LMO/KTO interface under 330-nm illumination at 5 K (see **Fig. R1d**).

Therefore, the giant enhancement of nonreciprocal transport is intrinsic, and it cannot be attributed to the inhomogeneous carrier distribution at the interface.

Fig. R4 | Homogeneity test and the 2D conducting characteristic of the 2DEGs at the CZO/KTO interfaces. **a**, The resistivity of the arbitrarily selected electrodes by applying both $I = \pm 40 \mu\text{A}$ in the dark and under 330-nm illumination, respectively. The temperature is 5 K without the applied external field. The inset is the sketch of this Hall-bar electrodes for the homogeneity test. **b**, The anisotropic MR as a function of the direction of the applied field at 5 K and 6 T for the CZO/KTO heterostructures. The inset shows the sketch of the anisotropic MR measurement.

Action: We have added Fig. R4 in Supplementary Information as new Fig. S4. In addition, we have inserted the sentence: “The homogeneous conducting channel is evidenced by the arbitrarily selected multi-electrode measurements (Supplementary Fig. 4a). The anisotropic MR further reveals its 2D conducting characteristic with the thickness of around 4.35 nm (Supplementary Fig. 4b).” (Line 129 on Page 6).

(6) One suggestion: the authors can further verify the nonreciprocal transport behavior with the conventional second harmonic measurement.

Reply: We are grateful for the reviewer’s professional suggestion. Indeed, the second harmonic measurements by applying AC currents can also evidence the nonreciprocal transport behavior. Here, in order to further confirm the light-induced giant enhancement of nonreciprocal transport at the CZO/KTO interface, we only show the special situation of the in-plane magnetic field fixed perpendicular to the current because we cannot simultaneously rotate the in-plane magnetic field under light irradiation for the moment. As seen from **Fig. R5**, the second-harmonic resistance ($R_{xx}^{2\omega}$) at about $\pm 0.7 \text{ T}$ under 330-nm illumination exhibits a giant increase with about 115 times as compared with that in the dark. While the first-harmonic resistance (R_{xx}^{ω}) under 330-nm illumination has 0.07 times as compared with that in the dark. Thus, the nonreciprocal transport coefficient under 330-nm illumination also has the giant enhancement of three orders of magnitude.

Fig. R5 | Second harmonic measurements for the nonreciprocal transport of KTO-based 2DEGs. $R_{xx}^{2\omega}$ as a function of in-plane magnetic field under 330-nm illumination and in the dark at 5 K.

Action: We have added Fig. R5 in Supplementary Information as new Fig. S8. In addition, We have inserted the sentence with “**In addition, we also perform the second harmonic measurements to further confirm the light-induced giant enhancement of nonreciprocal transport at KTO (111)-based interfaces (Supplementary Fig. 8).**” (Line 206 on Page 8).

Thus, the manuscript should address the above questions for publication in Nature Communications.

Reply: In short, we greatly appreciate Reviewer #1 for raising these important and professional comments. We hope that our responses are sufficient to address the reviewer’s questions and the improved manuscript meets the high criteria of *Nature Communications*.

Report of Reviewer #2 (Remarks to the Author):

Please refer to the attached comments.

In “Light-induced giant enhancement of nonreciprocal transport at KTaO₃-based interfaces”, the authors used light irradiation to observe the enhancement of nonreciprocal transport in the Rashba SOC in 2DEG at the interface of superconducting and epitaxial CaZrO₃/KTaO₃ interfaces. With ultraviolet illumination, the authors observe an enhancement in the Rashba SOC effective field due to the larger Rashba coefficient and longer carrier relaxation time. The authors demonstrated the interplay of light and nonreciprocal transport. Transport properties were characterized through Hall bar measurements. The authors measured the WAL pattern and fitted it with the Maekawa-Fukuyama (MF) model to extract the Rashba SOC strength through

measured WAL patterns. This work is well structured and presents the significant experimental results clearly. However, there are quite a few major concerns to be addressed before it can be considered further for Nature Communications. Please see comments below:

Reply: We thank Reviewer #2 for his/her thorough review and positive comments on our work. The reviewer summarized our work very accurately and recognized our work with “*well structured*” as well as “*significant experimental results clearly*”. Now we provide the detailed point-by-point response to reviewer’s major concerns and minor comments below. We hope that our responses will meet the reviewer’s expectations.

Questions and comments:

Major concerns:

(1) *The authors only use a rather short passage (lines 70-78) on page 4 to introduce 2DEGs in a KTO-based interface. In my opinion, the authors should introduce more 2DEG properties in this interface and its connection to this work in a more in-depth manner. By doing so, readers who are unfamiliar with the KTO-based interface can better grasp the essence of this material and its applications.*

Reply: Thank you for this nice suggestion. We agree with the reviewer that the cutting-edge research on 2DEGs at KTO-based interfaces has recently aroused the extraordinary enthusiasm in condensed matter physics and interdisciplinary electronics due to the underlying profound physics and potential applications. We follow this valuable suggestion and make a more in-depth introduction on KTO-based 2DEGs.

Action: We have rewritten this paragraph, as reading:

“As a 5d transition-metal oxide, KTaO₃ (KTO)-based interfaces **hosting 2DEGs have aroused considerable interest owing to fascinating physical properties**, such as **2D anisotropic** superconductivity³¹⁻³⁴, tunable strong Rashba SOC³⁵⁻³⁹, and **robust** spin polarization⁴⁰. **Especially, the superconductivity at KTO-based interfaces is strongly dependent on the crystalline orientation of KTO with the highest superconducting transition temperature (T_c) of 2.2 K at (111)-oriented surface³¹, very different from SrTiO₃-based counterparts, indicating the underlying rich physics in KTO. Furthermore, the interfacial superconductivity can be further manipulated by the electric field with/without ionic-liquid gating, unveiling the electron-doped surface⁴¹. Specifically, KTO-based 2DEGs are also very sensitive to the conventional light source^{35,42,43}, providing an ideal platform to explore the complicated interplay **between photoelectrons and heavy 5d electrons, thus readily tuning the Rashba SOC strength in a broader range under light illumination**. However, the nonreciprocal charge transport via optical modulation in the KTO-based 2DEGs still remains elusive.”**

(2) *In lines 202-204 on page 8, the authors claim that there are close connections between the magnetoresistance (WAL effect), Hall effect, and nonreciprocal transport.*

Although there are relevant discussions in the following sections and some discussions in lines 266-271, it is quite difficult to grasp this “connection” and their underlying physics. For instance, in Figs. 2 and 3, the authors did point out that when the illuminated light varies from 400 nm to 300 nm, the WAL is enhanced significantly, but is then quickly suppressed. In the meantime, when a 330 nm light incident on the Hall bar, a nonlinear Hall resistance emerges due to the existence of two different carrier types. In addition, from Fig. 3 and 4, the authors suggest that there is a Lifshitz transition point in nonreciprocal transport coefficient, carrier density, and mobility. Are these three effects/observations connected or related in some way? Can they all be explained using the Ta-t_{2g} conduction band model mentioned in lines 251 and 268? The reviewer recommend that the authors should elaborate, or even present figures, to demonstrate the Ta-t_{2g} conduction band in the band structure and how the 330 nm optical pumping leads to electrons jumping from the valance band to the Ta-t_{2g} conduction band.

Reply: We are very grateful to the reviewer for bringing up this invaluable and inspiring comment. We can grasp the connection between WAL effect, Hall effect and nonreciprocal transport **through the criteria of $\gamma \propto \alpha_R \tau^2 / E_F$** . Based on the previous derivation in *Phys. Rev. Mater.* **4**, 071001 (2020), the strength of nonreciprocal transport is proportional to the Rashba coefficient (α_R) and the square of relaxation time (τ^2). In the dark condition, due to the low carrier density of n_1 , the Fermi level is low near the conduction band minimum (CBM) (**Fig. R6**). The Rashba spin splitting is relatively small ($\Delta = 35$ meV, **Fig. 4b**), leading to a small Rashba coefficient α_R . Besides, the low mobility also results in a short relaxation time τ . Thus, the nonreciprocal transport is weak in the dark. However, the 330-nm illumination makes a difference. Due to the significant excitation of the carriers from the KTO’s valence band maximum (VBM) to the CBM, the Fermi level moves upward and crosses the higher subbands (**Fig. R6**), leading to the emergence of the Lifshitz transition. Based on our DFT calculations (**Fig. 4a-c**) and prior works (*ACS Nano* **13**, 609-615 (2019), *J. Phys. Chem. Lett.* **13**, 2976-2985 (2022)), this process brings about an enhancement of Rashba SOC strength as well as the Rashba coefficient α_R , which is evidenced by the remarkable WAL effects. This increased Rashba SOC partly contributes to the enhancement of nonreciprocal transport. Besides, the second type of photocarriers occupying the higher subband has a much larger mobility under 330-nm illumination, which can be extracted from the Hall effect. This brings about a longer carrier relaxation time τ , also contributing to the enhancement of nonreciprocal transport. Consequently, it is a synergistic effect resulting from the increased Rashba SOC strength (WAL effect and Lifshitz transition) and prolonged relaxation time (Hall effect), as modulated by light, which eventually boosts a substantial enhancement of the nonreciprocal transport. The close connections can all be explained by the photocarrier excitation model containing the Ta-t_{2g} conduction band shown below, as kindly suggested by the reviewer (**Fig. R6**).

Fig. R6 | The physical mechanism of photocarrier excitation. In the dark, 2DEGs form at the CZO/KTO interface and there are abundant electrons in the potential well, as shown in Fig. 2c. Under the 400-nm illumination, electrons are excited from in-gap states of KTO to the CBM and transfer to the potential well. In this case, the intrinsic carriers increase a little and the second-type carriers emerge (grey arrows). Under the excitation of 330-nm light, abundant electrons are excited from the VBM to the CBM, inducing the considerable increase of both the intrinsic carriers and the second-type carriers as well as the E_F -level upward shift. However, under the excitation of 300-nm light, the total carrier density cannot be increased further. In this case, electrons can be excited to the higher subbands (above Band 3,4) of the t_{2g} conduction band. The higher-energy photoelectrons lead to increased interactions between electrons and the lattice (phonons), which results in a longer time for the electrons to return to a lower energy state. During this process, more photogenerated electrons and holes are likely to recombine in a non-radiative manner. Hence, compared to the condition under 330-nm illumination, the final carrier density decreases and E_F level moves downward.

Action: We have added Fig. R6 in Supplementary Information as new Fig. S12. We have optimized the discussion in “Theoretical analyses” section (Page 12-13). In addition, we have inserted the sentence with “**This phenomenon is attributed to the fact that the photocarriers tend to fill the Ta- t_{2g} CBM where the intrinsic carrier occupies, thus increasing n_1 remarkably (Supplementary Fig. 12).**” (Line 290 on Page 11).

(3) *In lines 227-229 on page 9, the authors present a hypothesis to explain the unexpected rise in the density when the incident wavelength is below 400 nm. Is this hypothesis based on any physical models or theories in the existing literature? The explanation isn't clear. For example, what is a two-band? A higher subband in the quantum well? Second, under light illumination, photo-induced carriers do not necessarily fill the ground state. The ratio of carriers in the excited and ground state would be determined by the pump (illumination) rate and carrier lifetime of transition from the excited state to the ground state. The authors shall discuss with more rigorous calculation to support their great discovery.*

Reply: Thank you for this in-depth discussion, which is a continued intimate comment related to the upper one (**Comment 2**). Our hypothesis is just based on the above physical models under the different excitation wavelengths shown in **Fig. R6**. Now we have updated the discussion according to this new figure. The reviewer is correct that the other band in the two-band model is the higher subband in the potential well, as indicated in **Fig. R6**. In response to the reviewer's second concern on photocarriers' transition between the excited state and ground state, we emphasize that most of photogenerated electrons would be trapped *by the potential well* (i.e., to increase the carrier density) and do not return to the ground state at VBM upon the continuous illumination. Thus, it is tentatively unnecessary to calculate the ratio of carriers in the excited and ground state since the whole photoexcitation process is a dynamical equilibrium.

(4) The explanation of the nonlinear Hall and WAL effect using light with wavelength shorter than 330 nm is not satisfactory. If the authors think bilayer (two-types of carriers) are a major concern for wavelength of 330 nm, why would it lead to the same (similar) results for 300 nm?

Reply: We thank the reviewer for bringing up this inspiring question. In fact, by fitting the Hall resistance at 300 nm using the two-band model (**Fig. R7b**), we believe that two-types of carriers also exist under 300-nm illumination in view of its nonlinear Hall characteristic, which is not evident from **Fig. 3a**. Thus, it would lead to the similar results compared to that at 330 nm. The difference is that the photocarriers excited by 300-nm irradiation leads to increased interactions between electrons and the lattice (phonons), which leads to a longer time for the electrons to return to a lower energy state. During this process, more photogenerated electrons and holes are likely to recombine in a non-radiative manner, resulting in a decrease in the total carrier density and ultimately an increase in R_s . This will further suppress the Rashba SOC strength, the carrier relaxation time and the light-induced nonreciprocal transport.

Fig. R7 | Hall curves perfectly fitted using the two-band model. a,b, The fitted curves of the Hall resistance at 330-nm and 300-nm illumination using the two-band model (equations (3) and (4) in the main text).

Action: We have added Fig. R7 in Supplementary Information as new Fig. S11b-c. We have updated the discussion in the main text (Line 173-180 on Page 7).

(5) *In line 152, the author applied in-plane magnetic fields, can they explain why? In WAL experiment, a perpendicular (to the 2DEG plane) magnetic field is applied. An in-plane would suppress the WAL effect (see F. E. Meijer et al., Phys. Rev. B, 70, 201307, 2004 or Phys. Rev. Lett., 94, 186805, 2005). Please the authors elaborate more the reasons behind their experiment concept.*

Reply: We appreciate your thoughtful comment regarding the field direction. Actually, we apply the in-plane magnetic field in nonreciprocal transport experiments but the perpendicular field in WAL experiments, respectively. According to the previous reports (see *Nat. Phys.* **13**, 578 (2017), *Nat. Commun.* **10**, 1540 (2019)), the nonreciprocal coefficient γ exhibits the maximum magnitude when the polarization (P), magnetic field (B), and the current (I) are orthogonal to each other in the case of polar systems. If we apply an out-of-plane field in the nonreciprocal transport experiments, the γ would become zero. Thus, it is reasonable to apply the in-plane magnetic field in the nonreciprocal transport experiments. Besides, we do agree with the reviewer's viewpoint that a perpendicular (to the 2DEG plane) magnetic field should be applied in WAL experiment because the in-plane field suppresses the WAL effect.

Action: We have rewritten the sentence in “**To further reveal the underlying physical picture**, we carried out MR and Hall-effect measurements by applying **an out-of-plane** magnetic field with the identical wavelengths.” (Line 228 on Page 9).

(6) *While gamma is defined in p. 4 line 171, but it is not clear why gamma is also proportional to another term defined in line 248 (p. 10). A proper explanation with a prior work is necessary.*

Reply: Thank you for pointing out this question on the gamma definition. On one hand, the gamma defined as $\gamma = \Delta R / (2BIR_0)$ is derived from the phenomenological equation (1). Through this definition, we are able to calculate gamma value from the experimental data. On the other hand, the criteria of $\gamma \propto \alpha_R \tau^2 / E_F$ stems from the underlying physical picture of the nonreciprocal transport. In the Rashba 2DEG systems, the presence of a current \mathbf{j} in the x direction will introduce an equivalent Rashba field $\mathbf{B}_j \sim \alpha_R (\mathbf{j} \times \mathbf{z})$ in the y direction due to the spin-momentum locking (see the response to **Comment 12, Fig. R10**). The field \mathbf{B}_j is added to (or subtracted from) the external field \mathbf{B}_{ext} , depending on the current direction, thus bringing about the nonreciprocal transport. Hence, the magnitude of the equivalent Rashba field \mathbf{B}_j determines the intensity of the nonreciprocal transport. A more detailed derivation of the criteria can be found in *Phys. Rev. Mater.* **4**, 071001 (2020).

Action: We have rewritten the sentence by citing a prior work (*Phys. Rev. Mater.* **4**, 071001 (2020)) with “Guided by the criteria of $\gamma \propto \alpha_R \tau^2 / E_F$ from the underlying physical picture of the nonreciprocal transport²³, it is evident that the increase in Rashba coefficient (α_R) contributes to the larger γ .” (Line 317 on Page 12).

(7) Fig. 1:

- (a): please label the atomic constituents of the CZO/KTO heterostructure. Also label the orientations.
- (b): It is difficult to see O atoms in the STEM image. The question is why the authors O at those specific locations? In one unit cell, there should be $3 \times$ O atoms (or multiples). It seems non-sense to only label only 1 O atom in (b).
- (d): The authors did not provide fabrication details of Hall bar devices. Second question, if a metal gate is deposited, how would light penetrate to the underlying heterostructure. Even though ultraviolet light might not be absorbed (or partially penetrate), the authors should provide the relevant argument and prior work. Third: the transition temperature for superconductivity is not abrupt. It is misleading to claim T_c is 0.43 K.

Reply: We greatly appreciate the reviewer for these invaluable comments on some of the figures.

- Fig. 1a: The atomic composition labels of “CZO” and “KTO” have been added near the heterostructure, and their orientations have also been labeled.
- Fig. 1b: It is indeed very difficult to see O atoms in the STEM image. To more clearly demonstrate the atomic arrangement, the atoms have been superimposed on top of the STEM image, similar to that in the literature (see *Science* **371**, 716 (2021)), and we have updated the figure with $3 \times$ O atoms in a unit cell.
- Fig. 1d: We have provided more fabrication details of Hall bar devices as follows. An amorphous LaMnO_3 (a-LMO) layer was firstly deposited on the (111)-KTO substrate at room temperature (**Fig. R8a,b**). Then, the photoresist is spin-coated onto the LMO surface and the Hall-bar pattern of photoresist is fabricated using a hard mask via UV exposure (**Fig. R8c,d**). After the selective etching by HCl/KI solution and removal of the photoresist, the LMO layer is patterned into a Hall bar geometry (**Fig. R8e**). The LMO-patterned KTO substrates were transferred to the PLD chamber again for the growth of CZO at high temperature. The growth of the CZO was performed at 750 °C with an oxygen background pressure of 1×10^{-5} mbar. Finally, the LMO-patterned CZO/KTO heterostructures were grown (**Fig. R8f**).

During the whole process of preparing CZO/KTO heterostructure, no metal gate was deposited and the electric contacts were made using ultrasonic aluminum wire bonding. When the heterostructure is exposed to the light, the absorption of light in the CZO layer is negligible due to its transparent property. So, the light can penetrate to the heterostructure, which is consistent with our prior work (see *Adv. Mater.* **35**, 2211612 (2023) and *J. Phys. Chem. Lett.* **13**, 2976 (2022)). Thus, the Hall-bar fabrication does not affect light penetration into the underneath heterogeneous structure.

Fig. R8 | Schematic illustration of the Hall-bar device fabrication process. a-b, The amorphous LaMnO_3 (a-LMO) film is deposited by the PLD on the KTO substrate. c, The photoresist is spin-coated onto the a-LMO surface. d, Hall-bar pattern of photoresist is fabricated by the ultraviolet exposure. e, The selective etching by HCl/KI solution and the removal of the rest photoresist, forming the Hall-bar pattern of the a-LMO film. f, The CZO layer is deposited by PLD. The 2DEGs thus forms at the a-LMO-patterned CZO/KTO interfaces.

We agree with the reviewer that the transition temperature (T_C) for superconductivity is not very abrupt. We have now replaced “sharp” by “rapid” (Line 133 on Page 6). However, we think that the definition of T_C (the temperature corresponding to 50% of the resistance) is fully consistent with the former reports (e.g., *Phys. Rev. Lett.* **126**, 026802 (2021), *Science* **371**, 716 (2021), *Science* **372**, 721 (2021)).

Action: We have updated Fig. 1 and the relevant discussion in the main text: “The enlarged atomic configuration overlapping on the STEM image is shown in Supplementary Fig. 1b.” (Line 118 on Page 5); “The detailed fabrication process of the Hall-bar device is presented in Supplementary Fig. 3.” (Line 126 on Page 5); “The T_C is determined to be 0.43 K, which is defined by the temperature corresponding to 50% of the resistance according to the previous work^{31,32,44}. Its T_C is much lower than those of non-epitaxial KTO-based heterointerfaces^{31,32}, but comparable to that of epitaxial LaVO_3/KTO (111) system (~ 0.5 K)⁴⁵.” (Line 137 on Page 6); “When the heterostructure was exposed to the light, the light absorption in the upper CZO layer was negligible due to its wide-bandgap property and transparent nature.” (Line 383 on Page 15). In addition, we have added Fig. R8 in the Supplementary Information as new Fig. S3.

(8) Fig. 2:

- (b): Please explain why there exists a hump of sheet resistance at wavelength of 350 ~ 450 nm? Then why under illumination with an even short wavelength, the sheet resistance is increased?
- (c): Please cite prior work on the in-gap defect states. Furthermore, it is better to show carriers can be excited under light illumination with its photon energy smaller than the bandgap of KTO. For example, carriers might be excited in the valence band at CZO side and jump to the minimum of the quantum well in the KTO. The required energy is smaller than the bandgap of KTO. Please comment on this.

Reply: Thanks for these insightful questions.

- Fig. 2b: We believe that the hump is attributed to the excitation of electrons from the KTO's in-gap states generated by oxygen vacancies. We have re-calculated the energy of the in-gap states and re-drawn the band structure diagram (**Fig. R9a**).

Second, we should clarify the increase of resistance with an even shorter wavelength. The photo-generated carriers under 300-nm illumination would be excited from the valence band maximum (VBM) to a higher subband due to the larger photon energy (4.14 eV). These photocarriers lead to increased interactions between electrons and the lattice (phonons), which leads to a longer time for the electrons to return to a lower energy state. During this process, more photogenerated electrons and holes are likely to recombine in a non-radiative manner, resulting in a decrease in the total carrier density and ultimately an increase in the sheet resistance. Similar results have been observed in our previous work (*Adv. Mater.* **35**, 2211612 (2023)).

- Fig. 2c: Thank you for your constructive suggestion. We have cited prior works on in-gap defect states (Line 149 on Page 6, Line 169 on Page 7). Besides, in other previous works (see *ACS Nano* **13**, 609 (2019); *Adv. Mater. Interfaces* **9**, 2200103 (2022)), light with photon energy lower than the KTO band gap was employed and it is found that the electrons in the KTO in-gap states could be excited to the conduction band. This is in good agreement with our observation (**Fig. 2b**). We also notice another assumption that carriers might be excited at the other side and jump to the potential well in the KTO (see *Adv. Mater. Interfaces* **7**, 2000646 (2022)). However, we think that this case unlikely occurs in our work. If the electrons at CZO side are excited to the potential well in the KTO, the holes left in the valence band at CZO side would contribute to the conduction, thus making CZO conductive. However, this is contradictory to our observation that CZO always keeps insulating under 330-nm illumination at 5 K (**Fig. R9b**).

Fig. R9 | Band structure diagram and the time evolution of the resistance of the CZO films under 330-nm illumination at 5 K. a, Schematic band structure diagram of the light-gating mechanism at the CZO/KTO interface. The thicker grey arrows denote the more photoexcited electrons. **b**, The resistance always maintains insulating ($\sim 10^7 \Omega/\square$) during the illumination, indicating that carriers would not be excited from the VBM at the CZO side.

Action: We have updated Fig. 2c and the relevant discussion in the main text: “The resistance at 5 K firstly decreases by $\sim 7\%$ upon 700-nm illumination, and then shows no significant change in the wavelength range of 700 to 450 nm due to the low photon energy (Fig. 2b), which largely determines the average location (~ 2.3 eV) of the existing in-gap state (Fig. 2c). Furthermore, a notable reduction of R_s is observed upon the exerted photon energy between 2.82 and 3.45 eV (i.e., $360 \text{ nm} \leq \lambda \leq 440 \text{ nm}$), which largely determines the average location (~ 3.1 eV) of the other in-gap state (Fig. 2c). The electrons residing in these two in-gap states generated from oxygen vacancies can be both excited to the conduction band minimum (CBM), thus increasing the conductivity^{50,51}.” (Line 161 on Page 7).

(9) Fig. 3(d): The two-band model seems unreasonable. If I look at the values of n_1 and n_2 , it is obvious the contributions from n_1 to the total density is dominant. This will make the total (effective) mobility close to μ_1 . However, Fig. 1f says another story. The mobility at low temperature is ~ 30 , which is way below μ_1 (100 ~ 200) at dark. Please explain.

Reply: Thank you for this meaningful question. We agree with the reviewer that the contributions from n_1 to the total density is dominant and the total (effective) mobility is close to μ_1 . However, we should point out that the mobility at low temperatures is $100\text{-}200 \text{ cm}^2\text{V}^{-1}\text{s}^{-1}$ (Fig. 1f), which is consistent with μ_1 at dark. We have added circles and arrows in Fig. 1f to show the data more clearly and avoid any misunderstanding. Thus, our two-band model fitting is reasonable.

Action: We have updated Fig. 1f by adding circles and arrows.

(10) Prior work not cited properly.

- *The authors mentioned the importance of their great results on spin-induced SOC effect for spin-orbitronics and rectification device applications, but did not address the potential how their results can be applied to practical device applications. The authors did not cite relevant reference on so-called rectification devices, which hampers readers to further recognize the impact of this work.*
- *In line 61~62 (p. 3), the authors claimed that the gate bias is limited to modulate the SOC strength? Please briefly explain and cite a prior work.*
- *The citation for the expression in equation (2) is missing. If the authors derived it, please provide the details in the supplementary part.*

Reply: We appreciate the reviewer for reminding us citation.

- We have added relevant references (*Nat. Commun.* **11**, 5634 (2020); *Nat. Commun.* **15**, 245 (2024)) on rectification devices (Line 64 on Page 3). We have also added other related references (refs. 7, 14 and 27) in Introduction section for the readability.
- We have cited a prior work with a brief explanation in the main text as “However, these electrical gating means remains limited to manipulate the Rashba spin-orbit coupling (SOC) strength and the Fermi level **due to the relatively weak carrier tunability**²², thus seriously restricting the γ enhancement.” (Line 68 on Page 3).
- We have added the citation for the expression in equation (2) (Line 251 on Page 10). And we have re-written this equation in a more reasonable style, which reads: “**The total magnetoconductance under the negligible Zeeman splitting can be written as**^{57,58}

$$\frac{\Delta\sigma(B)}{G_Q} = -\frac{1}{2}\Psi\left(\frac{1}{2} + \frac{B_i}{B}\right) + \Psi\left(\frac{1}{2} + \frac{B_i+B_{so}}{B}\right) - \ln\left(\frac{B_i+B_{so}}{B}\right) + \frac{1}{2}\Psi\left(\frac{1}{2} + \frac{B_i+2B_{so}}{B}\right) - \frac{1}{2}\ln\left(\frac{B_i+2B_{so}}{B_i}\right) - \frac{AB^2}{1+CB^2}.$$

Minor comments:

(11) *In line 253 on page 10, the authors mentioned Fig. 2h, but it is missing. Did the authors mean Fig. 3e?*

Reply: Yes, we mean Fig. 3e. We have corrected this in main text (Line 324 on Page 13).

(12) *Please briefly introduce how Rashba SOC effects contribute to non-reciprocal transport.*

Reply: The nonreciprocal transport can be connected with Rashba SOC by taking into account the locking $\alpha_R(\mathbf{k} \times \mathbf{z}) \cdot \boldsymbol{\sigma}$ between spin $\boldsymbol{\sigma}$ and momentum \mathbf{k} in a Rashba 2DEGs (**Fig. R10a**). In the presence of a current \mathbf{j} , the shift of the Fermi contour occurs, resulting in the nonequilibrium energy (**Fig. R10b**). This can be described by the

introduction of a magnetic field $\mathbf{B}_j \sim \alpha_R(\mathbf{j} \times \mathbf{z})$ acting on the spin σ . The field \mathbf{B}_j is added to (or subtracted from) the external field \mathbf{B}_{ext} , depending on the current direction (Fig. R10c). Thus, the nonreciprocal transport takes place in cross terms between \mathbf{B}_j and \mathbf{B}_{ext} .

Fig. R10 | Schematic illustration of the connection between nonreciprocal transport and Rashba SOC. **a**, Schematic of the electronic and spin structure of a Rashba 2DEGs system. Arrows represent the electron spin. **b**, Schematic of the shift of a Fermi contour by a bias current \mathbf{j} , generating a nonequilibrium spin density. **c**, Effective magnetic field \mathbf{B}_{eff} as a result of the current-induced magnetic field \mathbf{B}_j (with its sign depending on the polarity of the applied current) and an external field \mathbf{B}_{ext} .

(13) The authors did not provide the light power or intensity, which is important for the quantitative analysis.

Reply: The light power density is set at $\sim 0.3 \text{ mW cm}^{-2}$ for all applied wavelengths to avoid any light-induced heating effect. This important information has been added (Line 385 on Page 15).

(14) In line 105 (p. 5), the EDX data only suggest there exists a hetero-interface, but not a good indicator for the abruptness. I would recommend the authors do not use “abrupt”. Instead, “clear” is probably a suitable term to describe their data.

Reply: Thank you for your kind suggestion. We have now replaced “abrupt” by “clear” (Line 120 on Page 5).

(15) In line 112 (p. 5), the transition of resistance does NOT drop to zero abruptly, so using “sharply” isn’t appropriate.

Reply: Yes, the word “sharply” cannot accurately describe the resistance transition. We have now replaced “sharp” by “rapid” (Line 133 on Page 6).

(16) While the terms of inelastic scattering (H_i) and SOC fields (H_{so}) are defined in equation (2), it is better to briefly introduce the physical meaning of those two terms.

Reply: Following equation (2), we define the physical meaning of B_i and B_{so} . We have added the sentence as “**These two characteristic fields are introduced to characterize B -dependent quantum correction based on the inelastic scattering and SOC scattering processes.**” (Line 255 on Page 10).

(17) For the equation (2), G_0 should be corrected to $2e^2/h$ or $e^2/(\pi \times \hbar)$. ρ and $\Delta\rho$ needs to be clearly defined, too.

Reply: We have corrected G_0 to be $2e^2/h$ in the main text. And we are confused about the r and $\Delta\rho$. Maybe the reviewer means $\Delta\sigma(B)$ and $\sigma(0)$? We add the definition as “**We present the magnetoconductance $\Delta\sigma(B) = \sigma(B) - \sigma(0)$ in the unit of the quantum conductance ($G_Q = 2e^2/h$) under different wavelengths, where $\sigma(0)$ is the magnetoconductance at zero field (Fig. 3b).**” (Line 247 on Page 10).

(18) What is R_0 ? (Line 158). Please define.

Reply: We have added the sentence with “**where the ΔR represents the difference in resistance upon opposite directions of the currents and R_0 represents the longitudinal resistance at zero field.**” (Line 193 on Page 8).

(19) Discussion must be corrected to “conclusion” since there is NO discussion at all.

Reply: We have changed the section heading from “Discussion” to “Conclusion” (Line 340 on Page 13).

Report of Reviewer #3 (Remarks to the Author):

By means of light irradiation, Zhang et al. reported a giant enhancement of nonreciprocal transport in the Rashba 2D electron gases at superconducting and epitaxial $\text{CaZrO}_3/\text{KTaO}_3$ (111) interfaces. The nonreciprocal transport coefficient undergoes a giant increase with three orders of magnitude up to $10^5 \text{ A}^{-1}\text{T}^{-1}$. Furthermore, a strong Rashba spin-orbit coupling (SOC) effective field of 15 T is achieved with abundant high-mobility photocarriers under ultraviolet illumination, which accounts for the giant enhancement of nonreciprocal transport coefficient. The first-principles calculations further disclose the stronger Rashba SOC strength and the longer relaxation time in the photoinduced carrier excitation process, bridging the light-property quantitative relationship. Their work provides a new pathway to boost nonreciprocal transport in noncentrosymmetric systems and facilitates the promising applications in rectification devices and spin-orbitronic devices.

The manuscript has grown $\text{CaZrO}_3/\text{KTaO}_3$ (111) 2D electron gases and achieved a superconducting transition temperature of 0.43 K. In addition, ultraviolet irradiation was used to modulate nonreciprocal transport, achieving a significant enhancement of

nonreciprocal transport. Compared with modulation methods of other systems, the $\text{CaZrO}_3/\text{KTaO}_3$ (111) 2D electron gas in this manuscript has the high nonreciprocal transport coefficient, which is a good innovation. However, the manuscript lacks a certain data fitting process and derived formulas, and the theoretical part is not specific enough. I think more detailed explanations and improvements in details are needed. Without clarifying these key issues, the paper unfortunately did not meet the criterion of *Nature Communications* at the current stage.

Reply: We extend our sincere gratitude to Reviewer #3 for his/her comprehensive summary of our work and recognized the primary novelty of “*the high nonreciprocal transport coefficient*” as compared with modulation methods of other systems. The reviewer had concerns about the data fitting process, derived formulas and the theoretical part. In the below, we carefully address the key issues raised by the reviewer in a point-by-point manner. We hope that our improved manuscript has now reached the reviewer’s reservations as well as the high criterion of *Nature Communications*.

The specific issues are as follows:

(1) *The first thing I want to know is, is there any magnetic hysteresis in the MR-H and R_{xy} -H curves under light irradiation?*

Reply: Thank you for pointing out this insightful question. Upon careful examination of our experimental data, we do not observe any magnetic hysteresis in the MR-H and R_{xy} -H curves under light irradiation (**Fig. R11**). This indicates that light irradiation in our experiments cannot induce any spin-polarized 2DEGs.

Fig. R11 | Hysteretic MR and R_{xy} measurements. a,b, MR and Hall curves under 330-nm irradiation with the applied out-of-plane magnetic field swept back and forth. We cannot observe any discernible hysteresis in both MR and Hall curves, indicating that light irradiation in our experiments cannot induce any ferromagnetic/spin-polarized 2DEGs at CZO/KTO interfaces.

Action: We have added Fig. R11 in Supplementary Information as new Fig. S9. In addition, we have inserted the sentence with “**We do not observe any hysteresis in the MR and Hall curves under 330-nm irradiation (Supplementary Fig. 9), indicating that light irradiation cannot induce any spin polarization in KTO-based 2DEGs.**” (Line 240 on Page 10).

(2) Line-105: *The manuscript states that there is no elemental interdiffusion in Figure 1c, but there is obvious diffusion of Ca in KTO, which is inaccurate.*

Reply: Thank you for bringing up the concern regarding the elemental interdiffusion. Actually, there is no Ca diffusion into the KTO substrate. The observed Ca signal in **Fig. 1c** is the artifact because of the very close energy edges of Ca and K (**Fig. R12**). Furthermore, we do not detect any Ca signal in KTO from the EDX spectrum (**Fig. R12b**).

Fig. R12 | EDX elemental mapping and the corresponding spectrum. **a**, EDX elemental mapping at the interface. A quantitative spectroscopic analysis is conducted in the region within the blue boxed area. The white dashed line is a guide to indicate the interface. The scale bar is 5 nm. **b**, The corresponding EDX spectrum, where the characteristic peaks of K, Ta and O elements are clearly seen. However, no peaks of Ca element are detected, indicating the noticeable Ca element in the KTO substrate in **a** is actually an artifact due to the very close energy edges of Ca and K from EDX.

Action: We have added Fig. R12 in Supplementary Information as new Fig. S2. In addition, we have inserted the sentence with “**Note that the noticeable Ca element in the KTO substrate and K element in the CZO film (Fig. 1c) are actually an artifact due to the very close energy edges of Ca and K from EDX (Supplementary Fig. 2).**” (Line 122 on Page 5).

(3) Line-131: *The manuscript does not specify whether the light intensity (power) is consistent at different wavelengths.*

Reply: We apologize for ignoring the light power density. We have added this important

information in Methods section.

Action: We have added the sentence “The light power density was maintained a constant value of $\sim 0.3 \text{ mW cm}^{-2}$ for all the wavelengths to avoid any light-induced heating effect” (Line 385 on Page 15).

(4) Line-134: *When the wavelength of light is less than 330 nm, please explain why R_s increases as the wavelength decreases? (Figure 2b) Supplementary material S3 was not cited in the main text, nor did it explain the selection of temperature (measuring physical quantities such as R_s at $T=5 \text{ K}$).*

Reply: Thank you for this insightful question. We should clarify the increase of resistance with an even shorter wavelength. The photo-generated carriers under 300-nm illumination would be excited from the valence band maximum (VBM) to a higher sub-band due to the larger photon energy (4.14 eV). These photocarriers leads to increased interactions between electrons and the lattice (phonons), which leads to a longer time for the electrons to return to a lower energy state. During this process, more photogenerated electrons and holes are likely to recombine in a non-radiative manner, resulting in a decrease in the total carrier density and ultimately an increase in R_s . Similar results have been observed in our previous work (see *Adv. Mater.* **35**, 2211612 (2023)).

Besides, we appreciate the reviewer for highlighting this important question about the temperature dependence of nonreciprocal transport. We sincerely apologize for ignoring the Supplementary Material S3 in the main text. We have included additional discussions on temperature dependence in the revised manuscript with “Moreover, the nonreciprocal transport gradually weakens with increasing temperature and vanishes at around 40 K, which is attributed to the disruption of spin-momentum locking by quantum fluctuations (Supplementary Fig. 7).” (Line 203 on Page 8). We choose to conduct the tests at 5 K because the nonreciprocal transport is more pronounced at low temperatures. Additionally, maintaining temperature stability becomes challenging when sweeping the magnetic field at even lower temperatures due to the instrumental limitations.

Action: We have added the discussion on the origin of the R_s increase in the main text (Line 173-180 on Page 7). We have cited Fig. S7 in the main text (Line 203 on Page 8). In addition, we have also added the temperature selection in Methods section, which reads “As the nonreciprocal transport is more pronounced at low temperatures the temperature for the transport measurements was mainly set at 5 K.” (Line 391 on Page 15).

(5) Figure 2d: *Missing unit in y-axis MR(%).*

Reply: We have added the unit of MR in Fig. 2d.

(6) Figure 2c: What do the energies of -1.7 eV and -3.0 eV represent in the Schematic? What is the basis for obtaining these data?

Reply: Thank you for bringing this to our attention. According to the relationship between resistance and wavelength in **Fig. 2b**, two resistance plateaus are observed within the wavelength ranges of 360-440 nm and 450-700 nm. This suggests the presence of two distinct in-gap states with different energies in KTO. Based on the wavelengths at which the electrons of the two in-gap states are excited, these two in-gap states are corrected to be -3.1 eV and -2.3 eV, respectively.

Action: We have updated the discussion with “The resistance at 5 K firstly decreases by ~7% upon 700-nm illumination, and then shows no significant change in the wavelength range of 700 to 450 nm due to the low photon energy (Fig. 2b), which largely determines the average location (~2.3 eV) of the existing in-gap state (Fig. 2c). Furthermore, a notable reduction of R_s is observed upon the exerted photon energy between 2.82 and 3.45 eV (i.e., $360 \text{ nm} \leq \lambda \leq 440 \text{ nm}$), which largely determines the average location (~3.1 eV) of the other in-gap state (Fig. 2c).” (Line 162 on Page 7).

(7) Line-198: In this paper, the Hall resistance has two different slopes, which is used to determine the Hall effect behavior contributed by two types of carriers. There is a lack of specific fitting formula, image and fitting curve. In addition, there is no detailed explanation for the changes in carrier concentration n_1 and n_2 and mobility μ_1 and μ_2 obtained after fitting (Fig. 3d,e). For example, when the wavelength is less than 330 nm, the reasons for the decrease of n_1 , n_2 and μ_2 and the increase of μ_1 are not explained.

Reply: We have added the specific fitting formula (equations (3) and (4)) in the revised manuscript. Besides, we have presented the fitting curves at 300, 330 and 400 nm as follows (**Fig. R12**).

Fig. R13 | Hall curves perfectly fitted using the two-band model. a-c, The fitted curves of the Hall resistance at 400-nm, 330-nm and 300-nm illumination using the two-band model (equations (3) and (4) in the main text). Note that the kinks occurring at the Hall

resistance at around ± 1 T of 330-nm illumination in **b** are originated from the significant excitation of the second type of carriers with extremely high mobility at 330 nm.

The sudden decrease in carrier density ($n_{1,2}$) and mobility (μ_2) with 300-nm excitation (**Fig. 3d,e**) is also ascribed to the recombination of photogenerated electrons and holes, in good agreement with the R_s increase (**Fig. 2b**). Please see the response to **Comment 4**. The trivial increase in mobility (μ_1) with 300-nm excitation (**Fig. 3e**) is probably due to the decrease in intrinsic carriers' effective mass.

Action: We have added Fig. R13 in Supplementary Information as new Fig. S11a-c. We have added the fitting formula (Line 274 on Page 11) and discussion on the fitted results in the main text: “Nevertheless, a second type of carriers begins to emerge when $\lambda \leq 400$ nm. Remarkably, the carrier density (n_2) and mobility (μ_2) of the second-type carriers both undergo a giant increase at 330 nm to $0.3 \times 10^{13} \text{ cm}^{-2}$ and $15400 \text{ cm}^2\text{V}^{-1}\text{s}^{-1}$, respectively, which stems from the photoelectrons excited from the KTO's VBM under the significant light-excitation process (Fig. 2c).” (Line 282 on Page 11); “The sudden decrease in carrier density ($n_{1,2}$) and mobility (μ_2) with 300-nm excitation is also ascribed to the recombination of photogenerated electrons and holes, in good agreement with the R_s increase (Fig. 2b).” (Line 292 on Page 11).

(8) Figure 3b: Lack of interpretation of figure.

Reply: We have added the related interpretation and the figure caption of Fig. 3b in the main text “We present the magnetoconductance $\Delta\sigma(B) = \sigma(B) - \sigma(0)$ in the unit of the quantum conductance ($G_Q = 2e^2/h$) under different wavelengths, where $\sigma(0)$ is the magnetoconductance at zero field (Fig. 3b).” (Line 247 on Page 10) and in the figure caption “**b** The magnetoconductance curves in the dark and at different wavelengths. The solid red curves are the best fits to the MF model.” Line 479 on Page 20).

(9) Line-222: It is written that the second type of carrier appears at 330 nm. However, according to Figure 3d, when $330 \text{ nm} < \lambda < 400 \text{ nm}$, the concentration of the second type of carrier n_2 is not zero, indicating that the second type of carrier has appeared when the wavelength is just less than 400 nm.

Reply: Thank you for your comment regarding the appearance of the second type of carrier. Upon scrutinizing **Fig. 3d**, we agree with the reviewer's viewpoint that the second type of carrier has appeared when the wavelength is just less than 400 nm.

Action: We have adjusted the wording as “...Nevertheless, a second type of carriers begins to emerge when $\lambda \leq 400$ nm.” (Line 282 on Page 11).

(10) Line-224 and Figs.3d and 3e: How are Lifshitz transition points selected? (Figure 3c, d, e) What is the theoretical basis for the Lifshitz transition points? Or are these just

speculation? Are they related to the electronic band structure of KTO (111) in any way? Please clarify.

Reply: Thank you for pointing out these insightful questions on the Lifshitz transition points. A Lifshitz transition is related to the electronic band structure of KTO (111). It is revealed at a critical carrier density, above which the Fermi level starts to cross the upper subband and a second type of carrier begins to emerge (*J. Phys. Chem. Lett.* **13**, 2976 (2022)). Based on this theoretical foundation, we believe that our previous speculation of the Lifshitz transition point was inaccurate. Considering that the second type of charge carriers begins to appear when $\lambda \leq 400$ nm, we determine the Lifshitz transition point at 400 nm.

Action: We have rewritten the sentence with “The Lifshitz transition is thus believed to occur **at about 400 nm (Fig. 3d,e).**” (Line 286 on Page 11). And Fig. 3d,e has been also updated.

(11) Line-259: *The image reference is incorrect; it should be Figure 4e instead of Figure 4d.*

Reply: We have corrected this typo in the revised manuscript (Line 335 on Page 13).

(12) Figure 4a: *Compared with dark conditions, the electronic band structure will change under light conditions [see, e.g., Ref. 27]. This manuscript did not calculate the electronic band structure under two different conditions, nor did it indicate whether Figure 4a is the band structure in the dark or under light conditions. It is recommended to supplement the electronic band structure and the weights of d_{xy} , d_{xz} , and d_{yz} under the two conditions.*

Reply: Thank you for your insightful comment regarding the band structures under different conditions. We have calculated the electronic band structures of KTO under different Hubbard U values (3-5 eV). We find no significant differences among them (**Fig. R14**) due to the relatively weak electron correlation of the $5d$ transition-metal oxides. This is very different from SrTiO₃-based $3d$ -electron interfaces in our previous work (*J. Phys. Chem. Lett.* **13**, 2976 (2022)) where the electron correlation is very strong. Therefore, we believe that illumination does not markedly alter the band structures of KTO but rather induces the notable change of the Fermi level, thereby influencing the Rashba SOC strength as well as the nonreciprocal transport. In **Fig. R14**, we have provided the weights of d_{xy} , d_{yz} and d_{xz} (black, purple and blue circles) orbitals. Representative Fermi surfaces with $E_F = 0.03$ and 0.25 eV are displayed in **Fig. 4b,c**, which are marked with the dashed black and red lines in **Fig. 4a**, respectively. These two Fermi surfaces denote the dark and 330-nm illumination conditions, respectively.

Fig. R14 | The band structures from first-principles calculations plus Hubbard U . a-c, The calculated electronic band structures (Band 1-4, orange and green lines) and the weight of d_{xy} , d_{yz} and d_{xz} (black, purple and blue circles) orbitals of 12-Ta layers of KTO (111) surface for $U = 3, 4$ and 5 eV, respectively. The weight of the orbitals is expressed by the size of the corresponding circles. There is no significant difference among calculated electronic band structures of KTO under different Hubbard U values, indicating that light only impacts on the location of E_F .

Action: We have added Fig. R14 in Supplementary Information as new Fig. S13. We have also updated Fig. 4 and the relevant discussion in the main text: “**Note that there is no significant difference among calculated electronic band structures of KTO under different Hubbard U values (3-5 eV), suggesting that light only impacts on the location of E_F (Supplementary Fig. 13), thereby influencing the Rashba SOC strength as well as the nonreciprocal transport.**”.

(13) Figure 4e: What do the orange and green curves represent respectively? Please add which corresponds to A_{QMR} and which corresponds to relaxation time τ ?

Reply: We have added arrows to clearly indicate the physical quantity represented by each curve in the main text, as shown below (**Fig. R15b**).

Fig. R15 | QMR fitting. a, The QMR as a function of the in-plane magnetic field at various

wavelengths and in the dark. The solid red curves are the best fits according to equation (5) in the main text. **b**, A_{QMR} and the relaxation time (τ) as a function of wavelength. The corresponding parameters of the dark condition are also shown.

Action: We have also added the QMR fitting figure (Fig. R15a) in Fig. 4 and the related description for clarity: “The QMR is defined as $[\text{MR}(B, +I_x) + \text{MR}(B, -I_x)]/2$, and it is fitted by the following expression²³:

$$\text{QMR} = \frac{3}{4} \left(\frac{g\mu_B}{\hbar} \right)^2 \tau^2 B^2 \quad (5)$$

where $\frac{3}{4} \left(\frac{g\mu_B}{\hbar} \right)^2 \tau^2$ is defined as A_{QMR} , g is the g -factor, μ_B is the Bohr magneton, and \hbar is the reduced Plank constant. Since the A_{QMR} is independent on α_R and E_F , the τ can be deduced from A_{QMR} .”.

REVIEWER COMMENTS

Reviewer #1 (Remarks to the Author):

- 1) The Rashba SOC parameters deduced from the WAL fitting seem to be contradictory to the MR data. As shown in Fig. 3c and Fig. S10, the WAL fitting derived $B_{so}(L_{so}, \tau_{so})$ is smaller (larger) than $B_i(L_i, \tau_i)$ with light wavelength $\lambda \geq 500\text{nm}$, which corresponds to the weak localization scheme. However, the MR data in Fig. 3a show the typical weak antilocalization behavior with $\lambda \geq 500\text{nm}$.
- 2) It is necessary to mention the effective mass (m^*) value and its rationality when the author calculate the Rashba coefficient and spin-splitting energy.
- 3) The authors give a specious explanation on the sudden drop (increase) of the carrier density (sheet resistance) upon 300nm light illumination. I still doubt the non-identical photon flux might be one of the main reason for the sudden change, since the power density of light was kept constant and thus the light with a shorter wavelength obviously has a lower photon flux. The authors claim that "Therefore, for the each-wavelength excitation, we provide the strong enough photon flux, namely the photon numbers are much larger than those of excited electrons, ensuring the excitation of all electrons". I agree that the photon numbers are much larger than the numbers of excited electrons, but how could the authors ensure the excitation of all electrons? The photoexcitation process is a dynamical equilibrium, there should be more excited electrons with more illuminating photons, as evidenced by Fig. R2.

Reviewer #2 (Remarks to the Author):

The authors did respond most of my comments and questions properly. However, some major concerns have not been addressed properly. I recommend a major revision until the following concerns are addressed.

1. There are still some concerns on the response to my comment 2 in the previous version. In the captions of Fig. R6 (in response)/Fig. S12 (in the supplementary information), the authors described that under 300-nm light excitation, the number of carriers will not be increased due to the stronger interaction between those excited electrons and phonons, so they tend to lose energy with a much short relaxation time. Thus, electrons tend to recombine with holes. This explanation requires more evidence and further discussion. For example, the authors should compare the relaxation rates of electrons with phonons, recombination rate of electrons and holes, with the pumping rates of light excitation. If the electron number cannot be increased further, the relaxation and recombination rates need to be much faster than the optical pumping, which is difficult to believe. Under light illumination, a semiconductor usually becomes an intrinsic material since the numbers of the induced electrons and holes are more than the original carrier number (by doping). By only increasing the photon energy without changing (reducing) the light intensity, I wouldn't think their explanation is appropriate.
2. In the response to the comment 3, the reviewer misunderstood the ground state, which is in the conduction band (Not in the valence band). Since the system is constant pumped by light excitation, electrons in the valence band will be excited to the conduction band, where there exists more than one subband. Therefore, some electrons are pumped to the ground state or excited states in the conduction band. The relative number should follow a simple two-band model by modifying (using) the Boltzmann expression of $n_2/n_1 \sim \exp(-\Delta E/kT)$

under equilibrium. The authors thought it is unnecessary to know the numbers of electrons in those two subbands. However, their argument about Fig. R6/ Fig. S12 did discuss the electron populations under light with a wavelength ~ 330 nm and ~ 300 nm are different, which I strongly think requires the calculation of electron numbers in both states. Even more contradictory, the authors did reply in their response 3 “The reviewer is correct that the other band in the two-band model is the higher subband in the potential well,…” When they fit Fig. 7b, the authors did consider a two-band model for their new Fig. 7b. If they did not get the numbers of electrons in the two subband, how did they get the fitting results?

3. The authors gave two definitions of nonreciprocal transport coefficient γ but didn't connect them. I suggest they should place part of their reply (for example, see below) to the manuscript to avoid confusing readers.

“On one hand, the gamma defined as $\gamma = \Delta R / (2BIR_0)$ is derived from the phenomenological equation (1). Through this definition, we are able to calculate gamma value from the experimental data. On the other hand, the criteria of $\gamma \propto \alpha R \tau^2 / EF$ stems from the underlying physical picture of the nonreciprocal transport.”

Reviewer #3 (Remarks to the Author):

All my concerns have been answered seriously, and I have no further comments on the revised manuscript.

I therefore recommend it for publication as it is.

RESPONSE TO REVIEWERS' COMMENTS (NCOMMS-23-60568A)

Report of Reviewer #1 (Remarks to the Author):

(1) *The Rashba SOC parameters deduced from the WAL fitting seem to be contradictory to the MR data. As shown in Fig. 3c and Fig. S10, the WAL fitting derived B_{so} (L_{so} , τ_{so}) is smaller (larger) than B_i (L_i , τ_i) with light wavelength $\lambda \geq 500$ nm, which corresponds to the weak localization scheme. However, the MR data in Fig. 3a show the typical weak antilocalization behavior with $\lambda \geq 500$ nm.*

Reply: Thank you for pointing out the contradiction regarding the WAL fitting when $\lambda \geq 500$ nm. Upon the careful examination of our experimental data, we agree that the Rashba SOC parameters deduced from the WAL fitting when $\lambda \geq 500$ nm were inappropriate due to the neglected constraints. Now we re-fit the SOC parameters of B_{so} (L_{so} , τ_{so}), B_i (L_i , τ_i), Rashba coefficient (α_R) and spin-splitting energy (Δ) in a more reasonable manner, with the fitted results shown in **Fig. R1**.

Fig. R1 | The deduced SOC-related parameters at various wavelengths at 5 K. a-d, The wavelength-dependent Rashba SOC effective field (B_{so}) and inelastic scattering field (B_i) in **a**, spin relaxation length (L_{so}) and dephasing length (L_i) in **b**, spin relaxation time (τ_{so}) and inelastic scattering time (τ_i) in **c**, and spin-splitting energy (Δ) and Rashba coefficient (α_R) in **d**, respectively. The corresponding parameters of the dark condition are also included in **a-d** for comparison.

Action: We have updated Fig. 3c and Fig. S10b-d using Fig. R1a and Fig. R1b-d, respectively.

(2) *It is necessary to mention the effective mass (m^*) value and its rationality when the author calculate the Rashba coefficient and spin-splitting energy.*

Reply: We appreciate your thoughtful comment regarding the effective mass (m^*) value. We used the $m^* \approx 0.3m_0$ (where m_0 is the free electron mass) to calculate the Rashba coefficient and spin-splitting energy according to the angle-resolved photoemission spectroscopy experimental result of KTO (111) (see *Sci. Rep.* **4**, 3586 (2014)).

Action: We have added this important information in the main text with “**When calculating the Rashba coefficient and spin-splitting energy, the effective mass m^* takes $0.3m_0$ (where m_0 is the free electron mass) according to the previous angle-resolved photoemission spectroscopy (ARPES) experimental result⁶⁰.**” (Line 272 on Page 11). We have also cited ref. 60 in the references (Line 646 on Page 25).

(3) *The authors give a specious explanation on the sudden drop (increase) of the carrier density (sheet resistance) upon 300 nm light illumination. I still doubt the non-identical photon flux might be one of the main reason for the sudden change, since the power density of light was kept constant and thus the light with a shorter wavelength obviously has a lower photon flux. The authors claim that “Therefore, for the each-wavelength excitation, we provide the strong enough photon flux, namely the photon numbers are much larger than those of excited electrons, ensuring the excitation of all electrons”. I agree that the photon numbers are much larger than the numbers of excited electrons, but how could the authors ensure the excitation of all electrons? The photoexcitation process is a dynamical equilibrium, there should be more excited electrons with more illuminating photons, as evidenced by Fig. R2.*

Reply: Thank you for your insightful comment on the sudden drop of the carrier density upon 300-nm illumination. We agree that there are more excited electrons with more illuminating photons in a low power range. However, since the light intensity applied in our experiment is excessive to excite most of electrons in the valence band, the difference in the photon flux between two wavelengths (300 nm and 330 nm) should not be responsible for the sudden change of the carrier density. **Figure R2** shows the dependence of stabilized sheet resistance on the light intensity (P) under 300-nm illumination. The 100% power density of light is 0.3 mW cm^{-2} . When the light intensity is small (e.g., 5-20%), the stabilized sheet resistance decreases remarkably with increasing P , indicating the important role of photon flux in the excitation process. Nevertheless, as the light intensity reaches 100%, the stabilized sheet resistance is saturated, indicating that the dynamical equilibrium reaches a limit. Even if we continue to increase the P (photon flux), no additional electrons are excited. Thus, we can ensure the excitation of most of valence electrons that are excited with the 100% power density of 0.3 mW cm^{-2} . Besides, even if we apply the same photon flux as 330 nm (namely 110% light intensity at 300 nm), the sheet resistance keeps unchanged ($\sim 110 \text{ } \Omega/\square$, see

Fig. R2), which is still two times larger than that at 330 nm ($\sim 50 \Omega/\square$, Fig. 2b). Therefore, we can conclude that the non-identical photon flux between 300 nm and 330 nm is not the main reason for the sudden change of the carrier density (sheet resistance).

Fig. R2 | The dependence of stabilized sheet resistance on the light intensity upon 300-nm illumination. The 100% power density of light is 0.3 mW cm^{-2} .

Overall, we greatly thank Reviewer #1 again for these valuable comments to further improve the quality of our manuscript.

Report of Reviewer #2 (Remarks to the Author):

The authors did respond most of my comments and questions properly. However, some major concerns have not been addressed properly. I recommend a major revision until the following concerns are addressed.

Reply: We thank Reviewer #2 for his/her further helpful comments on our work. We feel sorry that our previous response did not fully clarify the reviewer’s concerns. Here, we again take a closer look at the reviewer’s concerns and provide the point-by-point response below.

(1) *There are still some concerns on the response to my comment 2 in the previous version. In the captions of Fig. R6 (in response)/Fig. S12 (in the supplementary information), the authors described that under 300-nm light excitation, the number of carriers will not be increased due to the stronger interaction between those excited electrons and phonons, so they tend to lose energy with a much short relaxation time. Thus, electrons tend to recombine with holes. This explanation requires more evidence and further discussion. For example, the authors should compare the relaxation rates of electrons with phonons, recombination rate of electrons and holes, with the pumping rates of light excitation. If the electron number cannot be increased further, the relaxation and recombination rates need to be much faster than the optical pumping, which is difficult to believe. Under light illumination, a semiconductor usually becomes*

an intrinsic material since the numbers of the induced electrons and holes are more than the original carrier number (by doping). By only increasing the photon energy without changing (reducing) the light intensity, I wouldn't think their explanation is appropriate.

Reply: Thank you for your insightful comment on the 300-nm light excitation. From the previous work (see *Phys. Rev. Lett.* **108**, 117403 (2012)), the pumping electrons' time is faster than 100 fs, while the recombination time and relaxation time is on the order of picosecond. Hence, our former explanation of “*These photocarriers leads to increased interactions between electrons and the lattice (phonons), which results in a longer time for the electrons to return to a lower energy state*” is reasonable. We now provide more evidence to rationalize it. As clearly evidenced from its MR curve (**Fig. 3a**, the MR of 300 nm, also **Fig. R3** shown below for convenience), during the enhanced relaxation process, more photogenerated electrons would experience a **notable competition between the weak localization (WL) and weak antilocalization (WAL)** upon the higher photon energy (4.14 eV, i.e., 300 nm) excitation in addition to the likely recombination of photogenerated electrons and holes. Thus, the total carrier density responsible for conductivity decreases and the sheet resistance (R_s) naturally undergoes an increase. Furthermore, the R_s increase is also directly related to the derived B_{so} value (the SOC strength) from the MF theory (see **Fig. R1a**). The larger B_{so} , the smaller R_s . Since our work mainly focuses on the light-induced nonreciprocal charge transport in KTO-based interfaces by utilizing **the conventional light source (xenon light)** instead of ultrafast laser pulses, we cannot get the carrier dynamics from xenon light currently. Such an ultrafast carrier dynamic process is beyond the capacity of xenon light as well as the scope of the current research, which is an excellent research point in our future work. In spite of the limited capacity of Xenon light, it is still superior over ultrafast lasers to readily induce the giant enhancement of nonreciprocal transport in 2DEGs at the KTO-based interfaces for the potential device applications.

Fig. R3 | The MR curves under 330-nm and 300-nm illumination under the applied out-of-plane magnetic field at 5 K. It is apparent that the weak antilocalization (WAL) effect becomes notably suppressed under 300-nm irradiation, indicating the more photo-generated electrons experience the back scattering, resulting in the R_s increase.

In summary, although we cannot obtain the relaxation rates of electrons with phonons, recombination rate of electrons and holes, and the pumping rates of light excitation due to the lack of ultrafast pulses of the Xenon light, we can still rationalize our explanation with the notably suppressed weak WAL effect and the derived SOC strength (B_{so}) from the MR curves.

Action: We have added the related weak localization discussion in the main text: “During this **enhanced relaxation** process, more photogenerated electrons **would experience a notable competition between the weak localization and WAL upon the 300-nm excitation (see the next section) in addition to the likely recombination of photogenerated electrons and holes, thus** resulting in a decrease in the total carrier density **responsible for conductivity** and ultimately an increase in R_s .” (Line 178 on Page 7); “It is noteworthy that the WAL effect and nonlinear Hall characteristic are simultaneously suppressed when the wavelength decreases to 300 nm, **and the competition between the weak localization and WAL becomes notable**, which is due to the same physical mechanism as the R_s increase (Fig. 2b).” (Line 243 on Page 10); “The sudden decrease in carrier density ($n_{1,2}$) and mobility (μ_2) with 300-nm excitation is also ascribed to **the notable competition between the weak localization and WAL as well as** the likely recombination of photogenerated electrons and holes” (Line 302 on Page 12). Also, we have added the description behind the B_{so} evolution “**Note that the λ -dependent B_{so} evolution is consistent with that of the λ -dependent R_s (Fig. 2b), indicating their inherent direct relation and the same physical mechanism.**” (Line 264 on Page 10). Furthermore, we have also updated Fig. S12 (the schematic mechanism).

(2) In the response to the comment 3, the reviewer misunderstood the ground state, which is in the conduction band (Not in the valence band). Since the system is constant pumped by light excitation, electrons in the valence band will be excited to the conduction band, where there exists more than one subband. Therefore, some electrons are pumped to the ground state or excited states in the conduction band. The relative number should follow a simple two-band model by modifying (using) the Boltzmann expression of $n_2/n_1 \sim \exp(-\Delta E/kT)$ under equilibrium. The authors thought it is unnecessary to know the numbers of electrons in those two subbands. However, their argument about Fig. R6/ Fig. S12 did discuss the electron populations under light with a wavelength ~ 330 nm and ~ 300 nm are different, which I strongly think requires the calculation of electron numbers in both states. Even more contradictory, the authors did reply in their response 3 “The reviewer is correct that the other band in the two-band model is the higher subband in the potential well,...” When they fit Fig. R7b, the authors did consider a two-band model for their new Fig. R7b. If they did not get the numbers of electrons in the two subband, how did they get the fitting results?

Reply: Thank you for bringing up this inspiring comment. Based on your suggestions, we propose a physical model to calculate the relative number of photo-excited electrons in the ground states (n_1) and excited states (n_2) (see **Fig. R4**). The n_0 represents the density of total excited electrons and E_i ($i = 1, 2$) denotes the energy separation between

electronic states. According to the previous work (see *Appl. Phys. Lett.* **105**, 171905 (2014)), the photo-excited electrons population decays to the edge of the high energy band 3,4 and subsequently to the lower band 1,2 as a consequence of electron-longitudinal-optical (LO)-phonon relaxation.

Fig. R4 | Schematic illustration of the excitation with 3.76-eV photons and the carrier relaxation process.

Consider the population and depopulation of the electrons in the excited states (n_2):

$$\frac{\partial n_2}{\partial t} = n_0 \frac{h\nu_{LO}}{E_1} (\tau_{e-LO})^{-1} - n_2 \left[\frac{h\nu_{LO}}{E_2} (\tau_{e-LO})^{-1} + \tau^{-1} \right] \quad (1)$$

where $h\nu_{LO}$ is the LO phonon energy, τ_{e-LO} is the time required to emit a single LO-phonon and τ is the total rate of the non-radiative recombination. The solution of equation (1) under equilibrium is:

$$n_2 = n_0 \left[\frac{h\nu_{LO}}{E_2 \tau^{-1}} (\tau_{e-LO})^{-1} + 1 \right]^{-1} \frac{h\nu_{LO} (\tau_{e-LO})^{-1}}{E_1 \tau^{-1}} \quad (2)$$

Since our work mainly focuses on the light-induced giant nonreciprocal charge transport in KTO-based interfaces by utilizing *the conventional light source (xenon light)* instead of ultrafast laser pulses, we can only estimate these parameters based on the previous works. The value of $h\nu_{LO}$ takes 8.8 meV and τ_{e-LO} takes 31 fs (see *Appl. Phys. Lett.* **103**, 151903 (2013)), while τ takes 1.4 ps (see *Phys. Rev. Lett.* **108**, 117403 (2012)). The E_1 is 60 meV and the E_2 is 70 meV under 330 nm illumination. The n_2/n_0 calculated from equation (2) equals 2.25%, indicating a small part of photo-excited electrons resides in the excited states (Band 3,4) while the most of them reside in the ground states (Band 1,2). This result is perfectly consistent with our two-band model fitting results. The value (n_2/n_0) is also close to that derived from our two-band model ($\sim 2.45\%$). As to the condition of 300-nm illumination, E_1 increases to 440 meV. Thus, the value of n_2/n_0 decreases to 0.3%, which is also confirmed by fitting results (Fig. 3d).

In response to how to get the fitting results without knowing the numbers of electrons in the two subband, we should state that the standard two-band model for fitting the Hall curves in our work has been widely utilized to extract carrier densities and mobilities in many works (e.g., *Phys. Rev. X* **6**, 031035 (2016), *Phys. Rev. Lett.* **118**,

256601 (2017), *ACS Nano* **13**, 609 (2019), *Adv. Mater.* **33**, 2102102 (2021), *J. Phys. Chem. Lett.* **13**, 2976 (2022), *J. Phys. Chem. Lett.* **14**, 8684 (2023)). We have already described the two-band model in detail in the main text (Line 281-289 on Page 11). Thus, the carrier densities derived from the two-band model in our work are quite reasonable and reliable.

(3) *The authors gave two definitions of nonreciprocal transport coefficient γ but didn't connect them. I suggest they should place part of their reply (for example, see below) to the manuscript to avoid confusing readers.*

“On one hand, the gamma defined as $\gamma = \Delta R/(2BIR_0)$ is derived from the phenomenological equation (1). Through this definition, we are able to calculate gamma value from the experimental data. On the other hand, the criteria of $\gamma \propto \alpha_{RT}^2/E_F$ stems from the underlying physical picture of the nonreciprocal transport.”

Reply: Thank you for this constructive suggestion. We have inserted the discussion on the connection of two definitions of nonreciprocal transport coefficient γ in the main text.

Action: We have moved “the criterion of $\gamma \propto \alpha_{RT}^2/E_F$ ” behind the analysis of $\bar{\alpha}_R$, and briefly added the difference of two definitions to avoid confusing readers, which reads “Thus, the $\bar{\alpha}_R$ contributed by all the bands increases nearly one order of magnitude as compared to that in the dark (Fig. 4d), which should contribute to the larger γ in view of the criterion of $\gamma \propto \alpha_{RT}^2/E_F$ ²³. Here, the nonreciprocal transport coefficient γ is derived from the underlying physical picture of the nonreciprocal transport²³ rather than the direct phenomenological equation (1)^{9,10}.” (Line 331 on Page 13).

Also, we have modified the sentence “The nonreciprocal coefficient $\gamma = \Delta R/(2BIR_0)$ is derived from the phenomenological equation (1), which is used as the main figure of merit to quantify the nonreciprocal transport.” (Line 211 on Page 9).

Report of Reviewer #3 (Remarks to the Author):

All my concerns have been answered seriously, and I have no further comments on the revised manuscript. I therefore recommend it for publication as it is.

Reply: We are grateful for the reviewer's recommendation of publication in *Nature Communications*.

REVIEWER COMMENTS

Reviewer #1 (Remarks to the Author):

The authors have addressed most of my comments. The one question left is about the neglected constraints for the WAL fitting in the previous version. Please reveal more details of the fitting and clarify the differences made in the current version.

Reviewer #2 (Remarks to the Author):

I have no further questions regarding this manuscript.

RESPONSE TO REVIEWERS' COMMENTS (NCOMMS-23-60568B)

Report of Reviewer #1 (Remarks to the Author):

The authors have addressed most of my comments. The one question left is about the neglected constraints for the WAL fitting in the previous version. Please reveal more details of the fitting and clarify the differences made in the current version.

Reply: Thank you for pointing out the final question about the neglected constraints. Considering the four fitting parameters (B_{so} , B_i , A and C , where A and C denote the fitted parameters for the ordinary magnetoconductance in equation (2)), the WAL fitting is quite sophisticated. In the previous version, we conducted the fitting with the neglected fact that the WAL effect dominates when $\lambda \geq 500$ nm according to the MR results, as also kindly pointed out by the reviewer. This corresponds to the constraint of $B_{so} > B_i$.

We have added the constraints in the current version. According to the previous works on the WAL fitting at KTO-based interfaces (see *ACS Nano* **13**, 609 (2019), *Phys. Status Solidi RRL* **17**, 2200441 (2023) and *IEEE Electron Device Lett.* **44**, 1987 (2023)), the value of B_i is ~ 0.1 - 0.3 T while the value of B_{so} is > 1 T in the WAL-dominated region. Thus, we add the constraints of $B_i < 0.5$ T and $B_{so} > 0.5$ T during the fitting process. The fitting results are in good agreement with the MR curves where the WAL effect dominates. The current version is in a more reasonable manner. The difference made in the current version is that the value of B_{so} is about ten times larger than that in the previous version when $\lambda \geq 500$ nm.

We appreciate the reviewer's effort and comments that helped us to significantly improve our manuscript.

Report of Reviewer #2 (Remarks to the Author):

I have no further questions regarding this manuscript.

Reply: We thank the reviewer for his/her recommendation for publication in *Nature Communications*.

REVIEWERS' COMMENTS

Reviewer #1 (Remarks to the Author):

I have no more questions.

RESPONSE TO REVIEWERS' COMMENTS (NCOMMS-23-60568C)

Reviewer #1 (Remarks to the Author):

I have no more questions.

Reply: We thank Reviewer #1 for a constructive peer review process and the recommendation for publication in *Nature Communications*.